# The multiple antibiotic resistance operon of enteric bacteria controls DNA repair and outer membrane integrity

Prateek Sharma[1], James R.J. Haycocks[1], Alistair D. Middlemiss [1], Rachel A. Kettles[1], Laura E. Sellars[1], Vito Ricci[2], Laura J.V. Piddock [2] & David C. Grainger[1]

The multiple antibiotic resistance (*mar*) operon of *Escherichia coli* is a paradigm for chromosomally encoded antibiotic resistance in enteric bacteria. The locus is recognised for its ability to modulate efflux pump and porin expression via two encoded transcription factors, MarR and MarA. Here we map binding of these regulators across the *E. coli* genome and identify an extensive *mar* regulon. Most notably, MarA activates expression of genes required for DNA repair and lipid trafficking. Consequently, the *mar* locus reduces quinolone-induced DNA damage and the ability of tetracyclines to traverse the outer membrane. These previously unrecognised *mar* pathways reside within a core regulon, shared by most enteric bacteria. Hence, we provide a framework for understanding multidrug resistance, mediated by analogous systems, across the Enterobacteriaceae. Transcription factors MarR and MarA confer multidrug resistance in enteric bacteria by modulating efflux pump and porin expression. Here, Sharma et al. show that MarA also upregulates genes required for lipid trafficking and DNA repair, thus reducing antibiotic entry and quinolone-induced DNA damage.

[1] Institute of Microbiology and Infection, School of Biosciences, University of Birmingham, Edgbaston, Birmingham B15 2TT, UK. [2] Antimicrobials Research Group, Institute of Microbiology and Infection, College of Medical and Dental Sciences, University of Birmingham, Edgbaston, Birmingham B15 2TT, UK. Correspondence and requests for materials should be addressed to D.C.G. (email: d.grainger@bham.ac.uk)

The *Escherichia coli* multiple antibiotic resistance (*mar*) locus was identified as a determinant for cross-resistance to tetracyclines, quinolones and β-lactams[1]. Widespread among enteric bacteria, the system confers clinically relevant antimicrobial resistance in *E. coli*[2–6]. The *mar* phenotype results from induction of an operon designated *marRAB*[7]. In wild-type cells, *marRAB* expression is stochastic and enhanced by many antimicrobial compounds[7, 8]. Conversely, in some clinical isolates, *marRAB* expression is constitutive; mutations prevent auto-repression by MarR[7, 9]. Induction of *marRAB* results in expression of MarA, a transcription factor that controls multidrug efflux and porin production[9–13]. Like all members of the AraC-XylS family, MarA uses a dual helix-turn-helix motif to bind non-palindromic DNA targets[14]. This DNA sequence, called the 'marbox', is conserved in Gram-negative bacteria[14, 15]. Proteins related to MarA, such as SoxS, Rob and RamA, also recognise the marbox and have overlapping regulatory effects[15–17]. However, interplay between these factors is complex. For example, Rob binds DNA with low specificity, but high affinity, and is usually sequestered in a non-functional state[18, 19]. Conversely, MarA and SoxS have a lower affinity for the marbox but form ternary complexes with RNA polymerase that enhance recognition of marbox-containing promoters[19]. It is likely that Rob behaves differently to other marbox binding proteins because Rob has a multimerisation domain and can interact with DNA using only one of its two helix-turn-helix motifs[18–20].

Models that depend on drug efflux and reduced porin production cannot explain all *mar* phenotypes[21, 22]. For example, substantial resistance to minocycline is retained in Δ*marR* cells lacking *tolC* or *acrAB*[22]. Such observations have stimulated attempts to define the complete MarA regulon. Transcriptome analyses identified hundreds of genes putatively controlled by MarA[23, 24]. However, only three genes were common to independent studies[16]. Similarly, while 10,000 copies of the marbox occupy the *E. coli* genome, most are non-functional[19, 25]. Consequently, only a handful of experimentally confirmed MarA binding sites are listed in the Ecocyc database (Supplementary Table 1)[26].

In this work, we have mapped genome-wide DNA binding by both MarR and MarA using chromatin immunoprecipitation and DNA sequencing (ChIP-seq). We show that the *mar* regulon is extensive and encompasses both DNA repair and lipid trafficking systems. These regulatory events allow the *mar* locus to combat quinolone-induced DNA damage and penetration of the outer membrane by antimicrobial compounds. Although previously unrecognised, these pathways reside within a 'core' *mar* regulon shared by many enteric bacteria.

## Results

### Genome-wide distribution of marRAB-encoded transcription factors.

We used ChIP-seq to map global DNA binding by the *marRAB*-encoded transcription factors and the RNA polymerase σ[70] subunit. Experiments were done using the enterotoxigenic *E. coli* strain H10407[27]. The strain shares 3766 genes with *E. coli* K-12 and has 599 additional genes encoded by 25 discrete chromosomal loci and 4 plasmids (p948, p666, p58 and p52). We expected to identify a primary *mar* regulon, shared by most *E. coli* species, and additional targets specific to toxigenic strains. The MarA, MarR and σ[70] binding profiles of are shown in Fig. 1a. In each plot, genes are illustrated by blue lines (tracks 1 and 2), MarA binding is in green (track 3), σ[70] binding is in orange (track 4) and MarR binding is in black (track 5). As expected, MarR bound only to the *marRAB* promoter while MarA and σ[70] bound at many loci. We used MEME to identify sequence motifs associated with the 33 MarA binding peaks. Only one

statistically significant (*E*-value < 1e−12) motif was found (Fig. 1b, top panel). The motif closely resembles known MarA binding sites (Fig. 1b, compare top and bottom panel). For all peaks we determined the distance to the nearest start codon and sorted these distances into 100 bp bins. The distribution of peaks among the bins is illustrated in Fig. 1c; MarA and σ[70] most frequently bind the 100 bp preceding the 5′ end of a gene. Of the 33 MarA binding peaks, 15 were within 150 bp of a binding peak for σ[70] (Fig. 1c, inset). To support our ChIP-seq analysis, we tested binding of purified MarA to DNA fragments overlapping all 33 peaks. As a control, we also tested five DNA fragments from elsewhere in the genome. All but one of the DNA sequences derived from MarA ChIP-seq peaks bound MarA in vitro (Supplementary Fig. 1a). However, there was no binding to any of the control sequences (Supplementary Fig. 1b). The ChIP-seq and in vitro DNA binding data are summarised in Table 1. Five MarA targets were specific to *E. coli* H10407. Of these targets, one did not bind MarA in vitro, four were within prophage remnants, and only one was near to the 5′ end of a gene. Note that our list of MarA targets will not include sites occluded by other proteins in vivo[28]. For example, most known marboxes preferentially bind SoxS or Rob; we did not expect to isolate these loci (Supplementary Table 1)[19]. Consistent with this, our ChIP-seq identified only the known marboxes with a high affinity for MarA (Supplementary Table 1).

### Phenotypic landscape of the mar regulon.

We focused on the 28 MarA binding sites shared with *E. coli* K-12. Implications for antibiotic resistance were assessed using phenotypic profiling data (Fig. 1d; Supplementary Fig. 2)[29]. In the heatmap, rows represent *E. coli* strains lacking individual MarA-targeted genes. Columns indicate treatment with different antibiotics. Where columns and rows intersect boxes are coloured according to growth relative to wild-type cells. Hence, red boxes identify MarA targets required for innate resistance to the corresponding antibiotic. Of particular note are *xseA* and the *mlaFEDCB* operon. These encode the large subunit of exonuclease VII and a lipid trafficking ABC transport system, respectively[30–32]. The data indicate that *xseA* is a determinant for quinolone tolerance. Conversely, *mlaFEDCB* mediates sensitivity to several antibiotics including tetracyclines (Fig. 1d).

### PxseA is a MarA-activated promoter.

Figure 2a shows co-binding of MarA and RNA polymerase to the *xseA* locus. The sequence of the regulatory region is in Fig. 2b. The putative marbox (green) is immediately adjacent to the *xseA* promoter (P*xseA*; orange)[33]. To confirm binding at the marbox, we used electrophoretic mobility shift assays (EMSA) and DNAseI footprinting. For the EMSA experiments we used the *xseA*1 and *xseA*2 DNA fragments; the 5′ end of each fragment is denoted by an inverted triangle in Fig. 2b. The −36C mutation, present only in *xseA2*, ensures marbox inactivation. The EMSA experiment is shown in Fig. 2c. As predicted, MarA bound to the *xseA1* fragment (top panel) but not the *xseA2* fragment (bottom panel). In DNAseI footprinting, MarA protects the marbox from digestion and induces hypersensitivity at adjacent sites (Fig. 2d). The *xseA1* and *xseA2* fragments were also fused to *lacZ* in the reporter plasmid pRW50. Measurements of β-galactosidase activity suggest a role for the marbox in transcription activation (Fig. 2e). Consistent with this, raising the intracellular concentration of MarA stimulates P*xseA* only in the presence of the marbox (Supplementary Fig. 3). Note that this approach avoids deletion of *marA* and the possibility of compensatory regulation by SoxS or Rob.

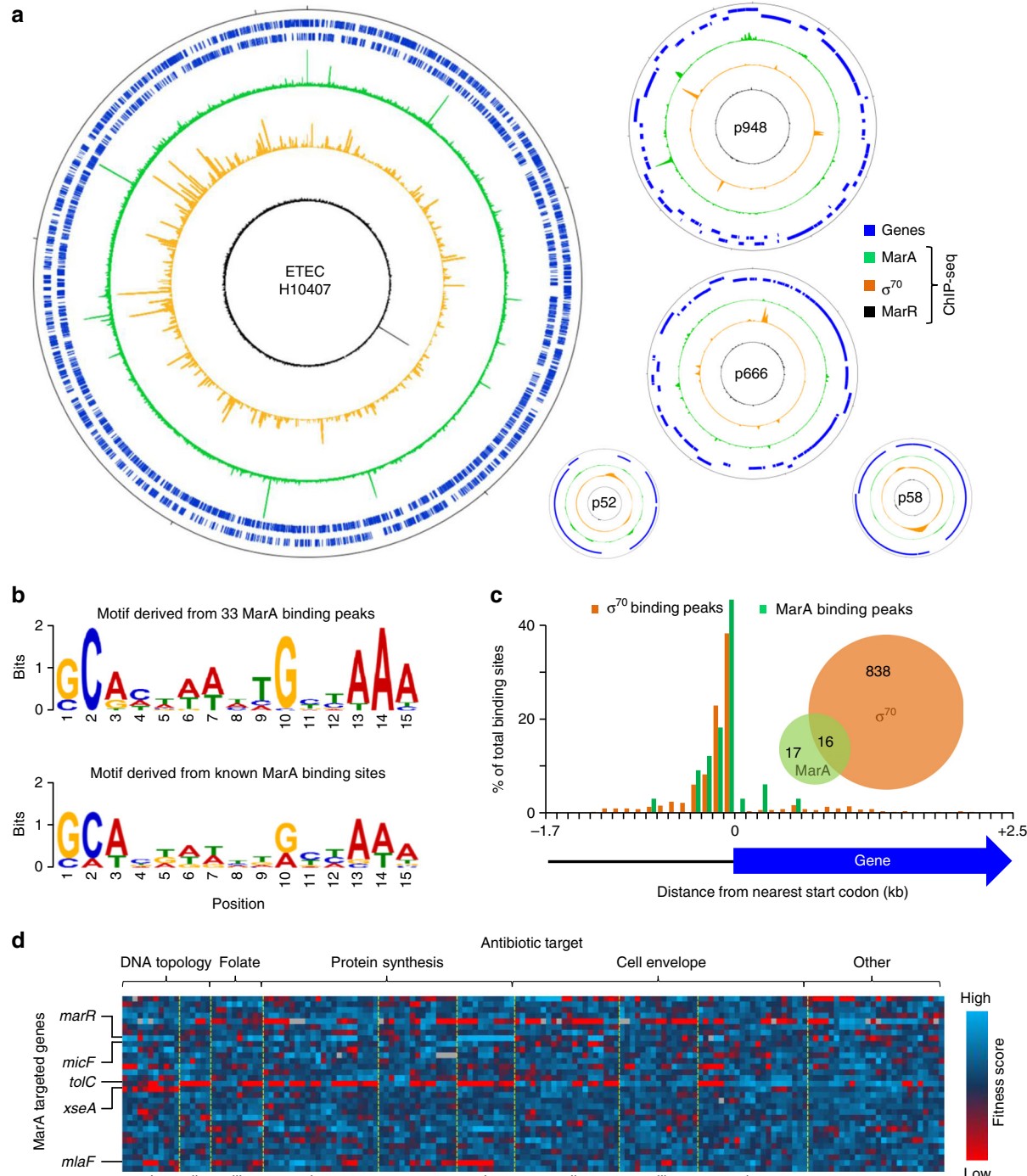

**Fig. 1** Global analysis of MarA and its target genes. **a** Genome-wide distribution of MarA, MarR and RNA polymerase in enterotoxigenic *Escherichia coli* strain H10407. Plots are shown for the H10407 chromosome and four plasmids. In each plot, the tick mark at the 12 o'clock position represents the first base pair (bp) of the DNA element. Subsequent tick marks are spaced by 1 Mbp (chromosome) 10 Kbp (p948 and p666) or 1 Kbp (p58 and p52). In each plot, tracks 1 and 2 (blue lines) show the position of genes, track 3 (green) is the MarA binding profile, track 4 (orange) is the RNA polymerase σ[70] subunit binding profile and track 5 (black) is the MarR binding profile. **b** DNA sequence motifs recovered from MarA binding peaks. The top panel shows a DNA sequence motif identified by MEME present in DNA sequences associated with MarA in ChIP-seq analysis. The bottom panel shows a DNA sequence motif generated by aligning experimentally verified MarA binding sites listed in Ecocyc. **c** Location of MarA and RNA polymerase binding peaks with respect to genes. A histogram depicting the distance between ChIP-seq binding peaks and the nearest 5′ end of a gene; data for MarA binding are in green and data for σ[70] binding are in orange. Each binding peak was allocated to a series of 100 bp bins. The inset is a Venn diagram that illustrates the number of MarA and σ[70] binding peaks that overlap. **d** Phenotypic landscape of the MarA regulon. The heatmap illustrates fitness scores[29] of strains lacking MarA target genes (*y*-axis) compared to the wild-type parent strain. Strains were grown in the presence of different antibiotics (*x*-axis). The antibiotics are clustered according to the cellular process targeted (labelled above heatmap). Drugs are further divided into classes *i* through *x* by yellow dotted lines. The classes are as follows: i quinolones, ii non-quinolone topoisomerase inhibitors, iii antifolates, iv macrolides; v aminoglycosides, vi tetracyclines, vii penicillins, viii cephalosporins, ix other cell envelope antibiotics, x miscellaneous. Individual row and column names are provided in Supplementary Fig. 2

**Table 1 MarA binding sites identified by ChIP-seq**

| ChIP peak[a] | MEME site centre[b] | Site sequence (5′–3′)[c] | H10407 genes[d] | MG1655 genes[e] | MarA binding in vitro[f] |
|---|---|---|---|---|---|
| *Chromosomal targets* | | | | | |
| **206** | 161 | gcacagacagataaa | ETEC0001 | *thrL* | +++ |
| **87300** | 87344 | gcacaattagctaat | ETEC0074<>ETEC0075 | *leuL<>leuO* | +++ |
| 184212 | 184180 | gcgttatctgttaat | ETEC0157 | *degP* | ++ |
| **428898** | 428876 | gcataaagtgtaaag | ETEC0400 | *lacZ* | ++ |
| **529310** | 529338 | gcacaaaatgacaaa | ETEC0500 | *ybaO* | +++ |
| 655542 | 655557 | gcactaaatgttaaa | ETEC0604 | *pheP* | +++ |
| **846408** | 846418 | ccacgcaaagctgac | ETEC0765<>acrZ | *modE<>acrZ* | ++ |
| 963300 | 963348 | cctatgagcgtaaaa | ETEC0889 | *ybiV* | + |
| 994036 | 994060 | gcattaattgctaaa | ETEC0916<>ETEC0917 | *grxA<>ybjC* | +++ |
| 1354006 | 1354020 | gcactaattgcaaaa | ETEC1264<>ETEC1265 | *ycgF<>ycgZ* | +++ |
| **1536992** | 1537038 | gcacaaattgtttaa | ETEC1438 | *fnr* | ++ |
| 1717096 | 1717112 | gcactaattgctaaa | ETEC1580 | *yneO* | +++ |
| **1739002** | 1739016 | ccacgtttttgctaaa | ETEC1599<>ETEC1600 | *marC<>marR* | +++ |
| **2321000** | 2321062 | gcactatttgctaaa | ETEC2157 | *yeeF* | +++ |
| **2538709** | 2538641 | gcactgaatgtcaaa | ETEC2344<>micF | *ompC<>micF* | ++ |
| **2727890** | 2727941 | gcattttttgctaaa | ETEC2509 | *ypeC* | +++ |
| 2755444 | 2755459 | gcaacaactgttaaa | ETEC2533><ETEC2534 | *yfeS><cysM* | +++ |
| **2887268** | 2887307 | gcattttttgcaaaa | ETEC2665<>ETEC2666 | *guaB<>xseA* | +++ |
| **3455714** | 3455708 | ccaatatccggcaaa | ETEC3200 | ETEC specific | - |
| **3569696** | 3569763 | gcacgtaacgccaac | ETEC3306<>ETEC3307 | *nudF<>tolC* | ++ |
| 3695690 | 3695710 | gcacaatctgcttac | (ETEC3426) | *(yhbV)* | +++ |
| **3733124** | 3733195 | ccagctttcgctaac | ETEC3460<>ETEC3461 | *mlaF<>yrbG* | +++ |
| **4289772** | 4289787 | gcacgaaacgttaaa | ETEC3977<>ETEC3978 | *ibpA<>yidQ* | ++ |
| 4348148 | 4348176 | gcacgatctgtatac | ETEC4032 | *mnmG* | ++ |
| 4494984 | 4495025 | ccgctttacggtaaa | (ETEC4151) | *(yihT)* | ++ |
| 4510208 | 4510184 | gcgcgttatgctgac | (ETEC4166) | *(yiiG)* | ++ |
| 4685066 | 4685041 | aggctaatcgtataa | (ETEC4304) | ETEC specific | +++ |
| 4686378 | 4686377 | ccaaaaacaggtaaa | (ETEC4307) | ETEC specific | ++ |
| 4737304 | 4737238 | gcaataaaagtcacg | ETEC4370<>ETEC4371 | *yjcB<>yjcC* | ++ |
| 5066076 | 5066105 | gcatcaaatgataac | ETEC4666<>ETEC4667 | *yjjP<>yjjQ* | +++ |
| 5093964 | 5093988 | ccgataaatgcgaaa | ETEC4702 | ETEC specific | ++ |
| **5132420** | 5132347 | gcaggaagcggcgaa | ETEC4739 | *deoB* | ++ |
| *Plasmid p948 targets* | | | | | |
| 65178 | 65159 | gcattttctgtcaaa | ETECp9480770 | ETEC specific | +++ |

[a]Genome coordinate of MarA ChIP-seq peak centre in H10407. Bold type indicates peaks within 150 bp of a σ[70] binding peak
[b]Genome coordinate of MarA binding site predicted by MEME
[c]Sequence of MarA binding site predicted by MEME
[d]Nearest gene to MarA binding site. Some MarA targets were between divergent (<>) and convergent (><) genes. Genes in parentheses indicate that the ChIP-seq peak is located within that gene
[e]*E. coli* K-12 homologues of ETEC genes in the previous column. *E. coli* K-12 MarA binding sites, listed in the Ecocyc database, are highlighted according to experimental confirmation (solid line) or prediction (dashed line)
[f]In vitro binding of purified MarA observed at a concentration of 0.3 μM (+++), 1.0 μM (++) or 1.7 μM (+) (Supplementary Fig. 1a). Note that five control DNA fragments (*cydD*, *ybiS* <>*ybiT*, ETEC2117, *cyoA* and P*estA*) did not bind MarA at any of these concentrations (Supplementary Fig. 1b)

**Activation of xseA mediates innate ciprofloxacin resistance.** Consistent with previous observations, the minimum inhibitory concentration (MIC) of ciprofloxacin decreased in cells lacking *xseA* (Supplementary Table 2). To determine if the ciprofloxacin hypersensitivity phenotype was dependent on both *xseA* and the marbox, we used genetic complementation. Hence, we constructed derivatives of plasmid pBR322 encoding *xseA* downstream of P*xseA*. This plasmid was able to rescue growth of an *xseA*::kan *E. coli* strain in the presence of ciprofloxacin but pBR322 with no insert was not (Supplementary Fig. 4). In the absence of ciprofloxacin, deletion of the marbox had little effect (Fig. 2f, top). However, in the presence of ciprofloxacin, the marbox was essential for growth (Fig. 2f, bottom).

**Innate ciprofloxacin resistance requires canonical XseA activity.** Exonuclease VII comprises large and small subunits; XseA and XseB, respectively. However, MarA binding was only detected upstream of *xseA* (Table 1). Hence, XseA could act independently of XseB to increase quinolone tolerance. Briefly, XseA consists of four functional regions: an N-terminal OB-fold, a central catalytic domain, a coiled-coil and a short C-terminal domain (Supplementary Fig. 5a)[34]. The OB-fold is responsible for DNA binding, the catalytic domain mediates exonuclease activity, the coiled-coil binds XseB and the C-terminal domain contains three α-helices[34]. To investigate the role of these activities, we introduced point mutations at key positions in *xseA* encoded by pBR322. The mutations, previously characterised by Poleszak et al.[34] are described in Supplementary Fig. 5a. Inactivation of any *xseA* determinant required for exonuclease VII function, including the interaction between XseA and XseB, results in hypersensitivity to ciprofloxacin (Supplementary Fig. 5b). Consistent with this, Nichols et al.[29] also concluded that both components of exonuclease VII were required for innate levels of ciprofloxacin resistance. We conclude that XseA acts via its described exonuclease VII activity in our assays.

**Cells lacking MarA-controlled xseA acquire DNA strand breaks.** Exonuclease VII is known to influence DNA repair[35, 36]. Hence, the simplest interpretation of our data is that MarA activates *xseA* to reduce DNA damage. We tested this prediction by visualising DNA damage in Hoechst-stained *E. coli* cells[37]. As expected, in the presence of ciprofloxacin, *xseA*::kan cells were

filamentous with abnormal nucleoid morphology (Fig. 2g, compare top two panels). Plasmid pBR322 encoding *xseA* was able to negate this phenotype, while deletion of the P*xseA* marbox blocked complementation (Fig. 2g, bottom two panels). We confirmed that the unusual nucleoid morphology was indicative of DNA fragmentation using pulse field gel electrophoresis (PFGE) (Fig. 2h). Chromosomal DNA ran as a tight band (lanes 1–3) but smearing was evident upon ciprofloxacin treatment of cells lacking *xseA* (lanes 4, 5). As in the microscopic analysis,

genetic complementation required an intact marbox (lanes 6, 7). For direct comparison, lanes 3, 5, 6 and 7 equate to the top through bottom panels in Fig. 2g, respectively.

**Transcription of the mla operon is activated by MarA.** We next turned our attention to the *mlaFEDCB* locus that also co-binds MarA and σ[70] (Fig. 3a). We confirmed binding of MarA to the *mlaFEDCB* regulatory DNA using EMSA and DNAse I

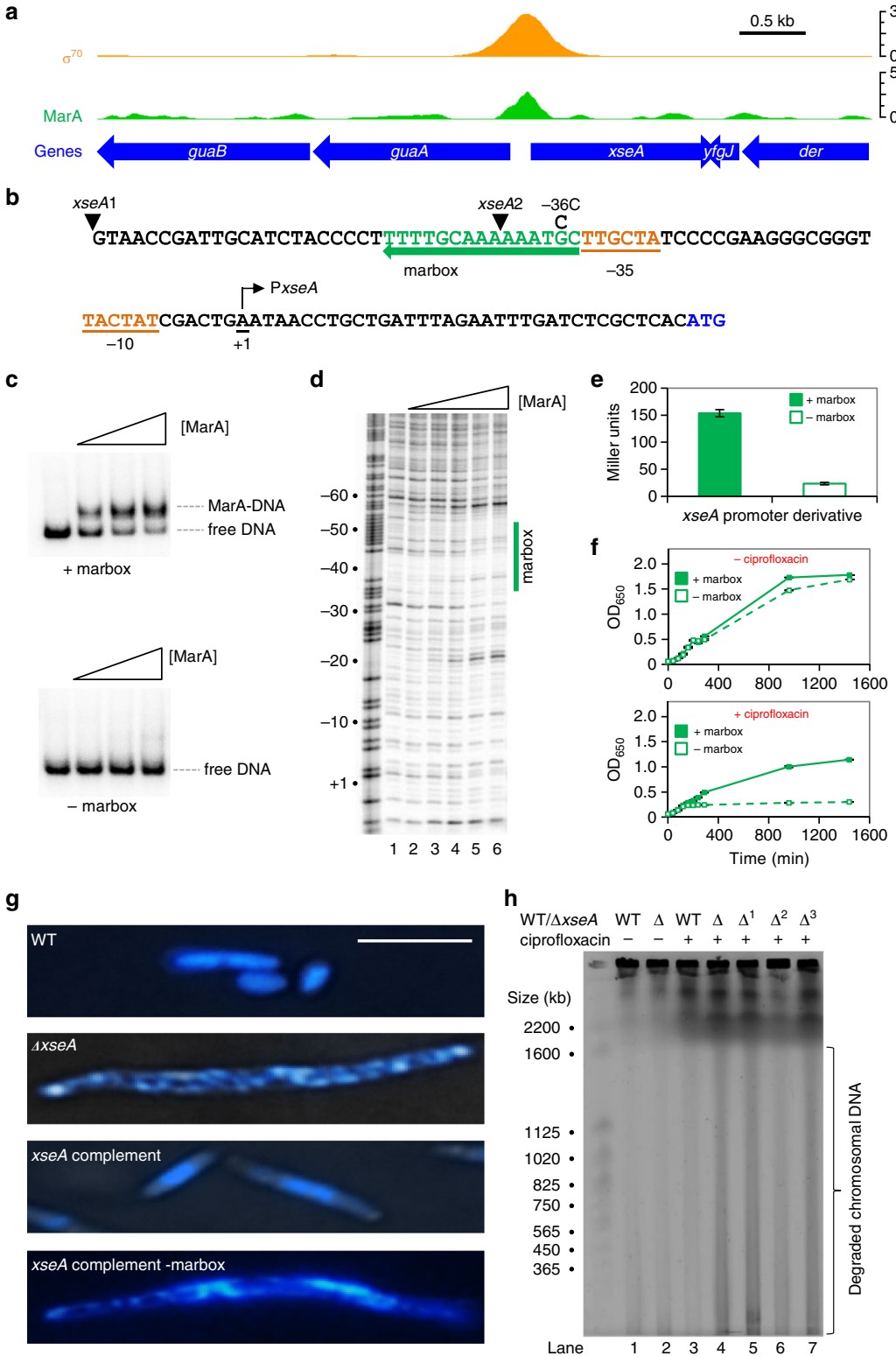

footprinting assays (Fig. 3c, d). The experiments were done using the *mlaF*1 and *mlaF*2 DNA fragments. The 5′ end of each fragment is marked by an inverted triangle in Fig. 3b. The results are consistent with MarA binding to the predicted marbox (green in Fig. 3b). However, the location of nearby promoters is unknown. To identify MarA regulated promoters, we used in vitro transcription assays. A DNA fragment containing the *mlaFEDCB* intergenic region was cloned upstream of the λ*oop* terminator in plasmid pSR. The resulting plasmid was used as a template for RNA synthesis (Fig. 3e). Note the 108 nt RNAI transcript is derived from the plasmid replication origin and serves as an internal control. Transcripts of 128, 148 and 157 nt in length were also observed (Fig. 3e). These initiate within the *mlaFEDCB* intergenic DNA at sites denoted by a bent arrow (Fig. 3b). Each messenger RNA start site maps downstream of promoter −10 and −35 elements. We refer to the promoters as *mla*P1, *mla*P2 and *mla*P3 (Fig. 3b). MarA activates *mla*P2 by binding adjacent to the −35 hexamer (Fig. 3b, e). Hence, deletion of the marbox reduces transcription derived from the *mlaFEDCB* intergenic region in *lacZ* fusion assays (Fig. 3f). Similarly, MarA overproduction activated *mlaFEDCB* only in the presence of the marbox (Supplementary Fig. 3). Inactivation of *mla*P1, which overlaps *mla*P2 and the marbox, increased the stimulatory effect of the marbox on *mla*P2 activity (Supplementary Fig. 6).

**Activation of mlaFEDCB mediates innate doxycycline resistance.** We quantified an eight-fold reduction in doxycycline MIC for *mlaE*::kan compared to wild-type cells (Supplementary Table 2). To confirm that doxycycline hypersensitivity was dependent on both *mlaFEDCB* and the marbox, we used genetic complementation. Thus, we constructed derivatives of plasmid pBR322 carrying *mlaFEDCB* and the upstream regulatory DNA. This plasmid was able to rescue growth of BW25113 *mlaE*::kan cells in the presence of doxycycline (Supplementary Fig. 7). As expected, deletion of the marbox had little effect in the absence of doxycycline (Fig. 3g, left). Conversely, in the presence of doxycycline, the marbox was essential for growth (Fig. 2g, right).

**Cells lacking mlaFEDCB have defective barrier function.** In Gram-negative bacteria, cell surface barrier function relies on outer membrane asymmetry; the inner and outer leaflets comprise phospholipids and lipopolysaccharides, respectively. The ABC transport system encoded by *mlaFEDCB* removes unwanted phospholipids from the outer leaflet[32]. Hence, *mlaFEDCB* could enhance drug resistance by improving outer membrane barrier function. Alternatively, ABC transport activity could support drug efflux[32]. To assess these hypotheses, we compared drug accumulation and efflux in wild-type and *mlaE*::kan cells.

Accumulation increased dramatically upon disruption of *mlaE* (Fig. 3h), but efflux was identical in both strains (Fig. 3i). Hence, our data are consistent with increased cell surface permeability but not defective efflux.

**Cells lacking mlaFEDCB have increased surface hydrophobicity.** Our model predicts reduced barrier function due to accumulation of phospholipids in the outer leaflet of the outer membrane. This should coincide with increased cell surface hydrophobicity. To test this, we measured partitioning of wild-type and *mlaE*::kan cells in a two solvent system (aqueous PUM buffer and p-xylene)[38]. In this assay, cells with increased surface hydrophobicity migrate to the organic phase. Consequently, turbidity of the aqueous phase is reduced. Consistent with this, *mlaE*::kan cells were retained in aqueous suspension less efficiently than the parent strain (Fig. 3j). Similar results were obtained in crystal violet adsorption assays[39]; the hydrophobic dye was bound more efficiently by the *mlaE*::kan strain (Fig. 3k). As expected, similar phenotypes arose in complementation experiments if the marbox was deleted in the pBR322 derivative carrying *mlaFEDCB* (Supplementary Fig. 8).

**SoxS and Rob binding to the xseA and mlaFEDCB promoters.** The SoxS and Rob proteins share 42 and 43% sequence identity with MarA across their helix-turn-helix determinants, respectively. Rob usually binds marboxes with a higher affinity than either MarA or SoxS[19]. Indeed, Rob can bind DNA fragments containing no marbox more tightly than MarA and SoxS bind canonical targets[19]. Hence, we next sought to better understand the comparative affinity of MarA, SoxS and Rob for the *xseA* and *mlaFEDCB* promoters. In control experiments, we confirmed that Rob had appreciable non-specific DNA binding activity (Supplementary Fig. 9a) and also bound a known marbox with higher affinity than MarA or SoxS (Supplementary Fig. 9b)[19]. Interestingly, most well-defined marboxes preferentially bind SoxS rather than MarA (Supplementary Table 1). This was not the case for the *xseA* or *mlaFEDCB* promoters, which bound MarA but not SoxS (Supplementary Fig. 10).

**The mar regulon is conserved among the Enterobacteriaceae.** Overexpression of MarA-like regulators is associated with clinically relevant resistance to quinolones and tetracyclines in many enteric bacteria[40]. Hence, we quantified conservation of MarA targets among the Enterobacteriaceae. The result of the analysis is illustrated in Fig. 4. As expected, MarA binding sites associated with efflux systems and porin expression were found in most species. The *xseA* marbox was similarly distributed. However, the best conserved marbox occurred upstream of *mlaFEDCB*; this

**Fig. 2** MarA binding upstream of *xseA* is important for DNA repair in the presence of ciprofloxacin. **a** ChIP-seq data for MarA and σ[70] binding to the *xseA* locus. Data have been smoothed in a 100 bp window. **b** DNA sequence upstream of *xseA* (start codon in blue) is shown. Relevant DNA elements are labelled and arrows indicate orientation. The *xseA* transcription start (+1) was identified by Davies and Drabble[33]. The 5′ end of the *xseA*1 and *xseA*2 DNA fragments are indicated by inverted black triangles. The *xseA*2 fragment carries the −36C mutation. **c** Electrophoretic mobility shift assays with the *xseA*1 fragment (+marbox) and the *xseA*2 fragment (−marbox). MarA was at a concentration of 0.3, 1.0 and 1.7 μM. **d** DNAseI footprinting experiment, using the *xseA*1 fragment, calibrated with a Maxam–Gilbert GA sequencing ladder. Positions relative to the *xseA* transcription start site (+1) are labelled. Concentrations of MarA are 0.3, 1.0, 1.7, 2.4 and 3.3 μM. The marbox is indicated by a green line. **e** Result of a β-galactosidase assay using lysates of JCB387 cells transformed with a reporter plasmid where *lacZ* expression is controlled by either *xseA*1 (+marbox) or *xseA*2 (−marbox). Error bars show standard deviation (*n* = 3). **f** The graph shows OD$_{650}$ values obtained for cultures of strain BW25113 *xseA*::kan grown in the presence or absence of 0.005 μg/ml ciprofloxacin. The BW25113 *xseA*::kan cells were transformed with pBR322 derivatives encoding *xseA* under the control of either the *xseA*1 fragment (+marbox) or the *xseA*2 fragment (−marbox). Error bars show standard deviation (*n* = 3). **g** Hoechst-stained BW25113 cells or the *xseA*::kan derivative. The term 'complement' denotes BW25113 *xseA*::kan transformed with pBR322 encoding *xseA* under control of the *xseA*1 fragment (+marbox) or the *xseA*2 fragment (−marbox). The scale bar is 5 μm and all panels are the same scale. **h** A pulse field gel electrophoresis experiment to analyse chromosomal integrity of BW25113 (WT) or the *xseA*::kan derivative (Δ). The *xseA*::kan derivative of BW25113 was transformed with either empty pBR322 (Δ[1]), pBR322 encoding *xseA* under the control of *xseA*1 fragment (Δ[2]) or *xseA*2 (Δ[3])

sequence was present in all but the most divergent *Cedecea neteri* genome. Overall, the *E. coli* MarA regulon is best conserved among *Escherichia* spp. and *Shigella* spp. However, a core regulon is shared by many Enterobacteriaceae. For example, *Salmonella*, *Citrobacter*, *Klebsiella*, *Enterobacter* and *Raoultella* spp. share a similar subset of MarA target genes including *tolC*, *micF*, *xseA* and *mlaFEDCB*.

## Discussion

MarA-like proteins are frequently implicated in the development of clinical resistance to quinolone and tetracycline family antibiotics[40, 41]. For example, constitutive MarA production reduces the rate of killing by norfloxacin and ofloxacin in *E. coli*[42]. This provides an opportunity for subsequent beneficial mutations to arise[4, 40]. Of the MarA target genes, we identified

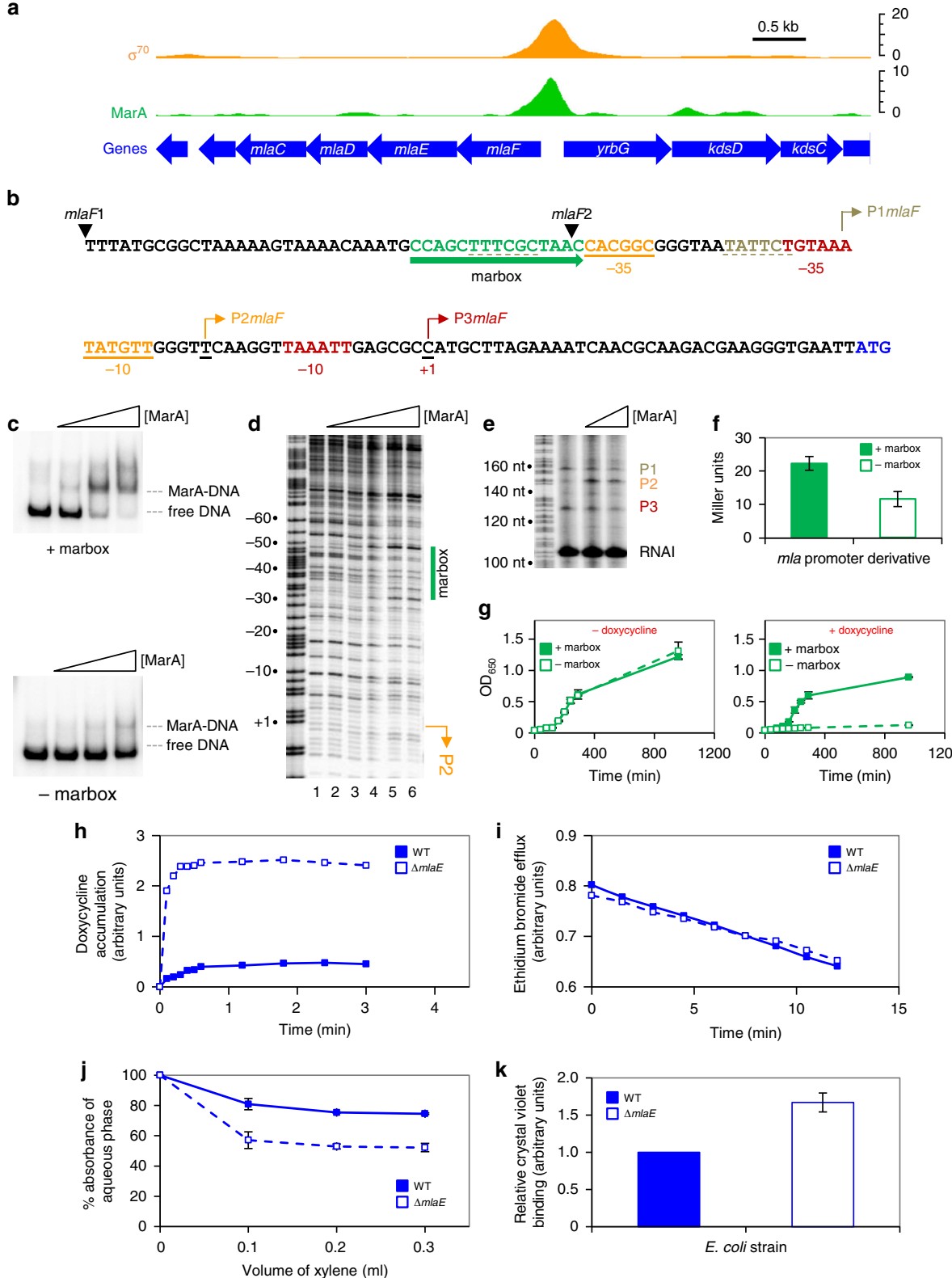

loss of *xseA*, rather than *tolC*, had the biggest impact on quinolone activity (Fig. 1d). Hence, we demonstrate a direct molecular link between MarA expression and reduced DNA damage (Fig. 2). This overlooked feature of the *mar* system in *E. coli* is likely to be widespread; the position and sequence of the marbox at P*xseA* is conserved in many enteric bacteria (Fig. 4). Deregulation of the *mar* operon is also associated with evolution of resistance to tetracyclines[43]. Indeed, clinically relevant levels of tetracycline resistance have been attributed solely to overproduction of MarA in *E. coli*[44]. Interestingly, substantial resistance to some tetracycline family antibiotics is retained in *marR* mutants lacking *tolC* or *acrAB*[22]. Hence, the *mar* regulon must encode other determinants for drug resistance. Our data show that the *mlaFEDCB* operon is of particular importance; the locus couples MarA to the control of lipid trafficking and outer membrane integrity (Figs. 1, 3). We note that tetracycline family antibiotics vary in their properties; doxycycline and minocycline are more hydrophobic than tetracycline[45]. Hence, increased surface hydrophobicity, due to inactivation of *mlaFEDCB*, renders cells most sensitive to the former two drugs (Fig. 1d). Consistent with our interpretation, others have noted a positive correlation between compound hydrophobicity and resistance provided by the *mar* system[46, 47].

The potential for cross-talk with other AraC family proteins complicates the study of gene regulation by MarA[19]. We show that both the *xseA* and *mlaFEDCB* promoters preferentially bind MarA rather than SoxS (Supplementary Fig. 10). Furthermore, upregulation of MarA expression activates these promoters even when SoxS and Rob are present (Supplementary Fig. 3). Even so, we do not exclude the possibility that MarA targets described here will bind closely related proteins in some circumstances (Supplementary Figs. 9, 10). Indeed, the *mlaFEDCB* locus is a target for RamA in *Klebsiella pneumoniae*[17]. Similarly, of 25 *E. coli* SoxS targets identified using ChIP-exo[48], 7 are bound by MarA in this study (*ybaO*, *acrZ*, *ybjC*, *ycgZ*, *micF*, *ypeC* and *yjcB*). Surprisingly, genes encoding transcription factors, including the global regulator FNR, were the second most common class of MarA targets (Table 1; Fig. 4). This provides an explanation for the pleiotropic effects of MarA on gene expression. Furthermore, because five MarA target genes were also a target for FNR, responses to oxidative and antibiotic stress must overlap[28, 49]. In conclusion, our work identifies previously unrecognised pathways to antibiotic tolerance mediated by MarA-like regulators. We suggest that the proteins involved are excellent drug targets; inhibition of XseA and MlaFEDCB should enhance the efficacy of quinolones and tetracyclines.

## Methods

**Strains, plasmids and oligonucleotides.** ETEC strain H10407 is described by Crossman et al.[27] The *E. coli* K-12 strains JCB387 and BW25113 are described by Page et al.[50] and Datsenko and Wanner[51], respectively. The *xseA*::kan and *mlaE*::

kan derivatives of BW25113 were obtained from the Keio collection[52] Plasmids pRW50 and pSR are described by Lodge et al.[53] and Kolb et al.[54] More detailed descriptions of strains and plasmids, along with sequences of oligonucleotides, are provided in Supplementary Table 3.

**Chromatin immunoprecipitation and DNA sequencing.** Immunoprecipitations with anti-MarA, anti-MarR and anti-$\sigma^{70}$ antibodies were done as described by Haycocks et al.[55] using lysates of bacterial strain H10407 transformed with plasmid pRGM9818 encoding *marA* under the control of the *tac* promoter[56]. This allowed us to circumvent repression of the chromosomal *mar* locus by MarR. Immunoprecipitations with anti-FLAG were done using lysates of H10407 transformed with pAMNF (encoding MarR-3xFLAG) or pAMNM (encoding MarR-8xMyc). Lysates were prepared from log phase cells cultured in LB medium. Anti-$\sigma^{70}$ and anti-FLAG were purchased from Neoclone (Madison, USA) and Sigma, respectively. Anti-MarA and anti-MarR were a kind gift from Lee Rosner and Bob Martin (NIH, Bethesda). Libraries were prepared using immunoprecipitated protein–DNA complexes immobilised with Protein A sepharose. DNA fragments were then blunt ended, A-tailed and barcoded. This was done using an NEB Quick Blunting and Ligation Kit, the Klenow fragment (5′–3′ exo-, NEB) and NEXTflex ChIP-seq barcodes (Bioo Scientific). Following elution of complexes from the Protein A sepharose, crosslinks were reversed and barcoded libraries were amplified by PCR. The number of PCR cycles was determined empirically for each library. After amplification, library concentration was quantified using Qubit analysis and real-time PCR. Equimolar library concentrations were pooled and sequenced using an Illumina MiSeq instrument. Sequencing reads are stored in ArrayExpress under accession number E-MTAB-5521 and E-MTAB-5591.

**Bioinformatic analysis of sequence reads.** The Fastq files obtained after DNA sequencing were converted into Fastq Sanger format, using FastqGroomer, and aligned to Genbank reference sequences (FN649414.1, FN649418, FN649417, FN649416 or FN649415) using BWA for Illumina. The reference sequences correspond to the H10407 chromosome and plasmids p948, p666, p58 and p52, respectively. The resulting SAM files were converted to BAM format using SAM-to-BAM. For each experiment, coverage per base was determined using multi-BamSummary. Subsequent processing was done using R. Data were normalised to the same average read depth and mean coverage per base was determined for each pair of replicates. The immunoprecipitations with anti-MarR were poorly efficient; enrichment of the *marRAB* promoter was evident but most peaks were associated with highly transcribed genes. These data served as a useful control for the anti-MarA data set; non-specific signals were removed by subtracting the anti-MarR signal from the equivalent anti-MarA value. Conversely, the anti-FLAG immunoprecipitations, with lysates of H10407 expressing MarR-3xFLAG, were highly efficient and isolated only the *marRAB* promoter. To select peaks for MarA or $\sigma^{70}$ binding, we used Artemis to generate a coverage plot and selected peaks scoring >2.7-fold (for MarA) or >3-fold (for $\sigma^{70}$) over background. A small number of peaks were called twice because they oscillated above and below the set threshold. Such duplications were removed manually. Four peaks for MarA binding were added manually after visual inspection. The peak centres were set as the centre of the region passing the cut-off rounded to the nearest integer.

**Bioinformatic identification of MarA binding sites.** After defining the MarA peak centres, we created a set of genome features in gff file format. Feature boundaries were 100 bp either side of each peak centre. The 201 bp DNA sequence corresponding to each feature was extracted using Artemis and submitted to MEME to search for motifs. The expected number of sites was set to one per sequence and the minimum motif width was set to 15 bp. A single statistically significant motif (*E*-value <1e−12) was recovered and this matched the known MarA binding consensus. The *E*-value is derived by MEME, from the motif's log likelihood ratio, taking motif length and background DNA sequence into account.

**Fig. 3** MarA controls outer membrane barrier function via activation of the *mlaFEDCB* operon. **a** ChIP-seq data for MarA and $\sigma^{70}$ binding at *mlaFEDCB*. Data are smoothed in a 100 bp window. **b** DNA sequence upstream of *mlaF* (start codon in blue). DNA elements are labelled and arrows indicate orientation. Transcription start sites identified in vitro are highlighted by bent arrows. The 5′ end of the *mlaF*1 and *mlaF*2 DNA fragments are indicated by inverted black triangles. **c** Electrophoretic mobility shift assays with the *mlaF*1 fragment (+marbox) and the *mlaF*2 fragment (−marbox). Concentrations of MarA are 0.3, 1.0 and 1.7 μM. **d** DNaseI footprinting experiment, using the *mlaF*1 DNA fragment, calibrated with a Maxam–Gilbert GA sequencing ladder. Positions relative to the *xseA* transcription start site (+1) are labelled. Concentrations of MarA are 0.3, 1.0, 1.7, 2.4 and 3.3 μM. **e** Results of in vitro transcription assays with the *mlaF*1 DNA fragment cloned in plasmid pSR. The gel is calibrated with a GA sequencing ladder. The RNAI transcript, derived from the pSR replication origin, acts as an internal control. Concentrations of MarA are 0.3 or 1.0 μM. **f** β-galactosidase assay with lysates of JCB387 transformed with a reporter plasmid where *lacZ* is controlled by either *mlaF*1 (+marbox) or *mlaF*2 (−marbox). Error bars show standard deviation ($n = 3$). **g** The graph shows $OD_{650}$ values for cultures of strain BW25113 *mlaE*::kan grown with or without 1.0 μg/ml doxycycline. The BW25113 *mlaE*::kan cells were transformed with pBR322 derivatives encoding *mlaFEDCB* under the control of either the *mlaF*1 fragment (+marbox) or the *mlaF*2 fragment (−marbox). Error bars show standard deviation ($n = 3$). **h, i** Accumulation of doxycycline or efflux of ethidium bromide as a function of time for BW25113 (solid line) or the *mlaE*::kan derivative (dashed line). **j** Percentage absorbance of the aqueous phase at equilibrium after mixing with p-xylene. Data are for BW25113 (solid line) and the *mlaE*::kan derivative (dashed line). **k** The graph shows crystal violet adsorption by BW25113 (solid bar) or the *mlaE*::kan derivative (open bar). Data are normalised relative to BW25113 cells. Error bars show standard deviation ($n = 3$)

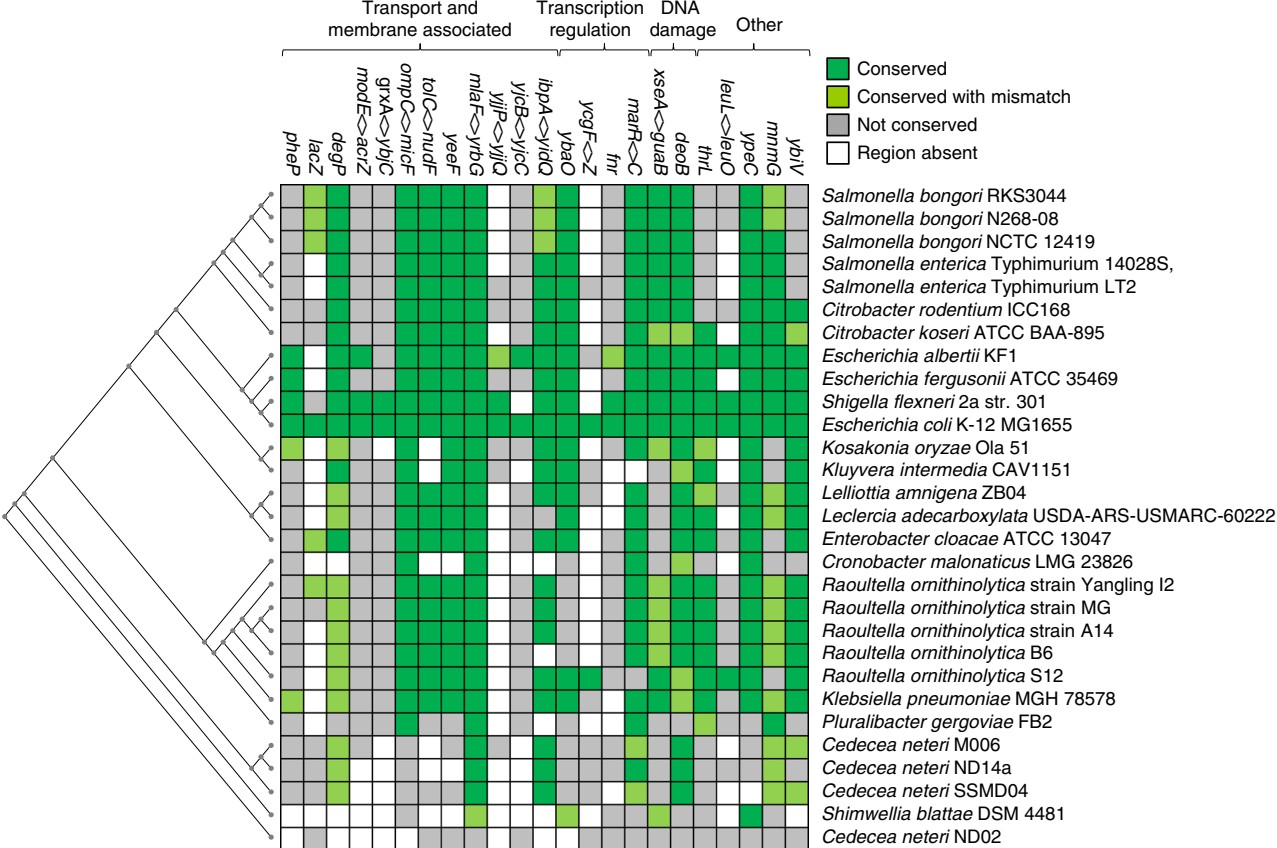

**Fig. 4** Phylogentic footprinting of the MarA regulon. The heatmap illustrates conservation of marboxes identified by ChIP-seq (x-axis) in the genomes of different enteric bacteria (y-axis). Dark green indicates conservation of a marbox with a maximum of one mismatch and light green indicates a maximum of two mismatches. Grey indicates that the intragenic region was identified but the marbox was poorly conserved or absent. Open boxes represent intergenic regions that were not identified in that genome. The evolutionary relationship between the different organisms, determined on the basis of the polA gene sequence, is indicated by a cladogram

To determine the distance between peak centres for σ70 and MarA, we used the fetch closest non-overlapping feature tool in Galaxy[57]. Peaks for σ70 and MarA described as overlapping were those centred were within 150 bp of each other.

**Phylogenetic analysis of the marbox.** Excluding *E. coli* species, we used BLASTp to search for genomes encoding MarA. We manually removed genomes encoding MarA with changes in the amino-acid sequence of either DNA recognition helix; this is indicative of altered DNA binding specificity. Many of the resulting 161 genomes were derived from closely related strains of the same species. These were omitted so that 29 representative genomes remained. Phylogeny was determined using the sequence of the *polA* gene from each organism and BLASTn pairwise alignments. We used BLASTn to search the 29 genome sequences for a match to the 201 bp DNA sequence derived from each MarA ChIP-seq peak. If multiple hits were obtained from a single genome, only the best match was used. If required, pairwise alignments were optimised manually to remove alignment gaps within marbox sequences. If no sequence match was identified, this was scored as 'region absent'. An identified marbox was scored as 'conserved' if it matched either the equivalent *E. coli* sequence, or the consensus marbox (5′-gcactaattgctaaa-3′) in at least 14 of the 15 possible positions. Sites were scored as 'conserved with mismatches' if the above criteria were satisfied at 13 of the 15 marbox positions. Sequences falling below this threshold we scored as 'not conserved'.

**Identification of H10407 MarA targets shared with K-12.** To locate regions of the *E. coli* K-12 genome equivalent to those in strain H10407, we used BLASTn. We compared the sequence of the marbox, intergenic region (defined as 200 bp upstream of the target gene or entire region between convergent genes) and target gene. Of the 28 shared marboxes, all but 6 were within identical intergenic regions. Of the six intergenic regions that differed, four contained a single base change and two had differences in just two positions. None of the intergenic regions contained insertions or deletions, consistent with identical juxtaposition of promoter elements and regulator binding sites between strains. Target genes were a minimum of 97% identical at the nucleotide level and 13 genes had 100% identity.

**Proteins.** Purified MarA, SoxS and Rob were a gift from Lee Rosner and Bob Martin (NIH, Bethesda). RNA polymerase was purified using a method derived from Burgess and Jendrisak[58]. Briefly, *E. coli* strain MG1655 was grown overnight in 3 l of LB broth. Cells were harvested by centrifugation and resuspended in 100 ml of lysis buffer (50 mM Tris-HCl pH 7.5, 150 mM NaCl, 2 mM MgCl₂, 0.1 mM DTT, 2 mM EDTA, 1 mM 2-mercaptoethanol, 5% glycerol, 0.2% Triton X-100 and 0.25 mg/ml lysozyme). One protease inhibitor cocktail tablet (Roche) was added per 20 ml of buffer. Cell lysis and DNA shearing was done using 4 × 30 s pulses, at 20% output, with a Misonix XL2020 tip sonicator. Lysates were cleared by centrifugation at 39,000×*g* for 45 min at 4 °C. Following filtration (0.45 µm filter) Polymin P and ammonium sulphate precipitation were done as described in Burgess and Jendrisak[58]. Precipitated protein was resuspended in TGED buffer (10 mM Tris-HCl pH 7.9, 5% glycerol, 0.1 mM EDTA and 0.1 mM DTT) containing 100 mM NaCl and passed through a HiPrep Heparin FF column (GE Healthcare). The column was washed with 0.1 M NaCl TGED and RNA polymerase was eluted in TGED using a gradient to 1 M NaCl. RNA polymerase containing fractions were pooled and protein precipitated using ammonium sulphate. After resuspension in TGED, RNA polymerase was further purified using a Mono Q HR column (GE Healthcare). Column washing and protein elution were as described in the previous step. RNA polymerase containing fractions were pooled and dialysed against −80 °C storage buffer (TGED, 0.1 M NaCl, 50% glycerol).

**DNA binding and in vitro transcription assays.** For EMSA experiments, DNA fragments were prepared using PCR as described by Shimada et al.[59] with oligonucleotides listed in Supplementary Table 3. Protein binding and subsequent electrophoresis were done as described by Chintakayala et al.[60] For footprinting experiments, DNA fragments were prepared as described by Grainger et al.[61] Protein binding, DNA digestion and electrophoresis were done as described by Singh and Grainger[62]. Briefly, DNA fragments were labelled at one end using [γ-³²P]-ATP and T4 polynucleotide kinase and used at a final concentration of ~10 nM in footprinting reactions. All reactions contained excess of herring sperm DNA (12.5 µg/ml) as a non-specific competitor. Our in vitro transcription assays were done as described by Haycocks et al.[55] Briefly, supercoiled pSR plasmid carrying promoter inserts (16 µg/ml) was pre-incubated with MarA in buffer containing 20

mM Tris pH 7.9, 5 mM MgCl$_2$, 500 μM DTT, 50 mM KCl, 100 μg/ml BSA, 200 μM ATP, 200 μM GTP, 200 μM CTP, 10 μM UTP and 5 μCi [α-32P]-UTP. Purified *E. coli* RNA polymerase was added to start reactions. DNAseI digested DNA and in vitro generated RNA transcripts were analysed on 6% DNA sequencing gels (molecular dynamics). The results were visualised using a Fuji phosphor screen and Bio-Rad Molecular Imager FX. Raw gel images are in Supplementary Fig. 11.

**β-galactosidase assays**. β-galactosidase assays were done as described previously[60, 63] using the protocol of Miller[64]. All assay values are the mean of three independent experiments with a standard deviation equivalent to <10% of the mean β-galactosidase activity. Cells were grown aerobically at 37 °C to mid-log phase in LB medium unless stated otherwise.

**Growth assays and MIC determination**. A single colony of each bacterial strain was used to inoculate 5 ml of LB broth that was incubated overnight at 37 °C. The OD$_{650}$ of overnight cultures was recorded so that equivalent OD$_{650}$ units could be used for sub-culturing each strain in fresh LB medium. The sub-cultures were then placed in a shaking incubator at 37 °C and, at 40 min intervals, 200 μl was transferred to a 96-well plate to measure OD$_{650}$ units. We used the MIC brothmic microtitre double dilution method[65] to assess antibiotic sensitivity. These assays were done in a final volume of 100 μl of LB medium in a 96 well round bottomed microtitre plate. Each well contained 10 colony forming units of *E. coli*, 100 μl of LB medium and antibiotics as required. The microtitre plate was covered with a sterile lid and kept in a gently shaking incubator for 24 h at 37 °C. The MIC was the lowest antibiotic concentration that prevented bacterial growth. Results were only accepted if the observed MIC for the control NCTC *E. coli* 10418 and ATCC *E. coli* 25922 strains was within one doubling dilution of the expected result.

**Microscopy**. Cells were grown in 1.5 ml of LB medium in the presence or absence of 0.005 μg/ml ciprofloxacin for 36 h at 37 °C. Cells were harvested by centrifugation and washed three times with PBS (140 mM NaCl, 2 mM KCl, 8 mM Na$_2$HPO$_4$, 1.5 mM KH$_2$PO$_4$). After washing, cells were resuspended in 10 μl of PBS containing Hoechst 33258 (5 μg/ml) and 40% (v/v) glycerol and left at room temperature for 10 min. Microscope slides were prepared by spreading 5 μl poly-L-lysine (10 mg/ml; Sigma) onto the centre of a slide. After drying, the slide was loaded with 5 μl of the cells suspension and a cover slip applied. The slides containing the cells were imaged using a Nikon Eclipse 90i microscope containing a Nikon Intensilight C-HGFI lamp, Hamamatsu ORCA ER camera (pixel size 6.45 μm) and Nikon Plan Apo VC ×100 Oil immersion lens (Numerical Aperture 1.4). The magnification was ×100 and a DAPI filter set allowed detection of Hoechst 33258-stained nucleoids. Exposure time was 90 ms, the excitation filter range was 340–380 nm and barrier filter range was 435–485 nm. Microscopy was done at room temperature and slides were imaged within 30 min of preparation. Images were analysed using the Nikon's NIS elements software.

**Pulse field gel electrophoresis**. Preparation of DNA for PFGE was based on the method described by Heath et al.[66] Overnight cultures of BW25113 and derivative strains were grown in presence or absence of 0.005 μg/ml ciprofloxacin at 37 °C for 36 h. After growth, cells were recovered, washed with PIV buffer (10 mM Tris-HCl pH 7.6, 1 M NaCl) and suspended in PIV at a final OD$_{650}$ of 1.7 units. The suspensions were incubated at 37 °C for 10 min and mixed with an equal volume of 1% PFGE grade agarose (Amresco Agarose LF) at 42 °C. Agarose plugs were prepared by pouring the suspension into PFGE moulds. After solidification, plugs were transferred to EC Lysis buffer (6 mM Tris-HCl pH 7.6, 1 M NaCl, 100 mM EDTA, 0.2% deoxycholate, 1% N-lauroylsarcosine, 1 mg/ml lysozyme, 20 μg/ml RNase) and incubated overnight in a gently shaking incubator at 37 °C. Buffer was removed and plugs were washed five times with TE buffer (10 mM Tris-HCl, pH 7.5, 1 mM EDTA). After washing, plugs were incubated overnight at 37 °C in ESP buffer (0.5 M EDTA pH 9.5, 1% N-lauryl sarcosine, 50 μg/ml proteinase K). The following day, plugs were washed five times with TE buffer and then stored in TE buffer at 4 °C until used. PFGE was done over 24 h in 0.5× TBE using a CHEF-DR II module system. The initial and final switch times were set to 60 and 120 s, the temperature was 14 °C, voltage was 6 V/cm and the included angle was 120°. The gel was stained using ethidium bromide.

**Accumulation and efflux assays**. Accumulation of doxycycline was measured using protocols derived from Mortimer and Piddock[67]. About 10 ml of fresh LB broth was inoculated with 250 μl of an overnight culture and placed in a shaking incubator at 37 °C. When the culture had obtained an OD$_{650}$ of ~0.7 units, the cells were harvested by centrifugation and resuspended in 10 ml of ice-cold PBS. After washing, cells were resuspended in PBS with a final OD$_{650}$ value of 20 units. The suspension was then transferred to a sterile universal container and left to equilibrate at 37 °C with magnetic stirring. Doxycycline was added to the cell suspension at a final concentration of 35 μg/ml. At timed intervals, after the addition of doxycycline, 500 μl of cell suspension was added to an Eppendorf tube containing 1 ml of ice-cold PBS. After mixing, cells were recovered by centrifugation at 4 °C. Samples were washed with ice-cold PBS to remove residual doxycycline and stored on ice until the end of the time course. After being resuspended in 1 ml of 100 mM glycine (pH 3), and incubated for 2 h at room temperature, cell debris was removed by centrifugation and fluorescence was measured. Efflux assays were based on protocols described by Blair et al.[68] Cells were initially grown as described above. After washing with PBS, cells were resuspended in PBS at an OD$_{650}$ of 0.3 units. Ethidium bromide was added to a final concentration of 5 μg/ml and the efflux inhibitor chlorpromazine was added to a concentration of 50 μg/ml[69]. The suspension was incubated at 25 °C for 60 min. Once loaded with ethidium bromide, cells were washed with PBS and resuspended in PBS containing 0.4% v/v glucose to induce efflux.

**Hydrocarbon and crystal violet binding assays**. Partitioning of cells between PUM buffer and p-xylene was done as described by Rosenberg et al.[38] Crystal violet binding assays were done as described by Halder et al.[39]

**Data availability**. ChIP-seq reads have been deposited in the ArrayExpress database under accession codes E-MTAB-5521 and E-MTAB-5591. The authors declare that all other data supporting the findings of the study are available in this article and its Supplementary Information files, or from the corresponding author upon request.

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

## Acknowledgements

We would like to thank Joe Wade, Mark Webber, Tony Maxwell and Natassja Bush for critical reading of the manuscript prior to submission and Ed Bevan for helping with PFGE experiments. We are greatly indebted to Bob Martin and Lee Rosner who have supported every aspect of the work by providing essential materials and advice; their input has been invaluable. This work was funded by BBSRC grant BB/N014200/1 awarded to D.C.G. and L.J.V.P. The Darwin Trust of Edinburgh, BBSRC and Wellcome Trust supported P.S., R.A.K. and A.D.M., respectively, with the award of a PhD studentship.

## Author contributions

P.S., A.D.M. and R.A.K. did the experimental work. J.R.J.H., L.E.S., V.R. and L.J.V.P. provided support with experimental work and participated in discussion. D.C.G. conceived the study and wrote the paper with input from all co-authors.

## Additional information

**Competing interests:** The authors declare no competing financial interests.

