## [Peer Review File · Nature Communications]

Reviewers' comments:

Reviewer #1 (Remarks to the Author):

The manuscript by Sharma et al describes results using ChIP-seq to identify MarA targets in enterotoxigenic *Escherichia coli* strain H10407. Subsequent experiments using in vitro DNA binding and transcription assays and in vivo antibiotic sensitivity and promoter reporter assays in *E. coli* K12 strains reveal two new targets of MarA; *xseA* that functions in DNA repair and *mfaFEDCB* operon that functions in outer membrane integrity. These functions were also shown to be important for greater resistance to ciprofloxacin and deoxycline, respectively. Although several previous studies have attempted to identify the MarA regulon using transcriptomics, these earlier results did not lead to a common consensus regulon and direct effects of MarA were not established. Here the new results show that MarA can directly bind many promoter regions and can mediate its effect of multiple antibiotic resistance by activating two new target genes.

The strength of the paper is discovering that MarA activates expression of *xseA* and the *mfa* operon, which increased resistance to certain antibiotics. Thus these results provide new insights into mechanisms of antibiotic resistance and the broader role of MarA. Overall, the authors present a compelling story but there are a few areas that require clarification.

1. The authors show that MarA directly binds these two promoter regions and that MarA can activate transcription of the P2 promoter of *mfa*. To demonstrate that MarA also regulated these promoters in vivo, the authors compared lacZ expression controlled by the upstream control region to one where the upstream control region was truncated to remove the MarA binding site. However since the authors also mentioned that these binding sites are the same as those bound by SoxS, Rob, and RamA, how did the authors rule out a role for these other transcription factors? This concern is reinforced by the fact that the ChIP-seq experiments were carried out in the enterotoxigenic *E. coli* strain but all of the in vivo assays were carried out with *E. coli* K12. Are the relevant promoter regions similar between these two types of *E. coli*? Were either *xseA* or the *mfa* operon induced by MarA in the previous transcriptomic studies?

2. The authors have concluded that resistance to deoxycline can be attributed to the action of MarA on the *mfaFEDCB* operon. Furthermore, the authors have shown that *mfaF* can be transcribed from three promoter elements (P1, P2 and P3). Based on the location of the marbox within the promoter region, they conclude that *mfaF* is transcribed primarily from P2. However the truncation of this promoter that they use to demonstrate a dependence on MarA also has the upstream promoter P1 deleted. Thus it is unclear how observed changes in expression can be concluded to be exclusively due to the action of MarA.

3. Previous studies have found that *dmsX* mediated activation of *marA* leads to multi drug resistance by ArcAB-TolC efflux pumps. However, the authors suggest an alternative explanation of multi-drug resistance but do not explain the discrepancy in results between these two studies.

Minor comments

1. It was not clear whether the levels of MarA play an important role in whether MarA exerts an effect at a given promoter and whether this is important in some of the discrepancies between previous genomic studies. Are the levels of MarA the same between the two *E. coli* strains?
2. Apparently only a few of the 12 confirmed MarA targets of *E. coli* K12 and reported in Ecocyc were found in the enterotoxigenic *E. coli*. Do the authors have an explanation?
3. Line 105. The spelling of electrophoretic.
4. Lines 115-122. Clarify whether it was previously known that *xseA* mutants are hypersensitive to ciprofloxacin.
5. Lines 126-136. Clarify whether strains lacking XseB are also hypersensitive to ciprofloxacin. What evidence allows you to conclude that XseA acts independently of XseB?
6. Line 256. The spelling of immunoprecipitated.

7. What is the mutation "-36C" in Figure 2b and what is its significance?
8. The legend for Fig 1 C does not describe adequately what the Venn diagram represents.

Reviewer #2 (Remarks to the Author):

NCOMMS-17-06813

Summary

The authors report a ChIP analysis in the pathogenic *Escherichia coli* strain H10407 of the binding sites of MarA: a major regulator driving naturally occurring multidrug resistance (MDR), likely hoping to find pathogen-specific roles for MarA. Finding only 1 such candidate, the authors follow up their hits in *E. coli* K-12. Interestingly, the authors recover only a small fraction of the validated *E. coli* Mar-box regulated genes in EcoCyc (last updated 2009, so likely incomplete). Although this is an incomplete global study, the authors do characterize two new Mar targets *xseA* (encoding a subunit of Exonuclease VII) and the phospholipid transfer system encoded by *mlaFEDCB* (previously identified as a target in *Pseudomonas*). This manuscript would be greatly improved by (1) greater transparency about the existing literature on marboxes and MarA-confirmed sites, (2) discussion of H10407-found MarA binding sites vs K-12 Mar boxes, and (3) evidence for the specific role of MarA at the defined sites (as opposed to the combined regulation of MarA/SoxS/Rob).

Major concerns

1. The authors seek to comprehensively map the regulon of MarR and MarA, in the pathogenic *E. coli* strain H10407, using ChIP using a *marA* or *marR* overexpression plasmid. Even though overexpression may overestimate binding sites, there is no evidence that their approach is anywhere near comprehensive. In this regard, it is important to determine whether the inability to identify previously validated *E. coli* Mar-controlled proteins derives from the fact that H10407 lacks Mar binding sites upstream of these genes or whether their approach was unable to detect these sites. This analysis will not only indicate whether their results are likely to be comprehensive; it will also indicate whether there is a great deal of variability in the Mar regulon between closely related organisms. At the very least, the authors could assess bioinformatically whether validated *E. coli* targets have Mar boxes in H10407. Additionally, it would be helpful in Table 1 to show K-12 EcoCyc confirmed MarA regulated genes that were not found by the authors' experiment.

2. Phenotypic characterization of the gene deletion phenotypes are a nice addition to the ms. However, there are several issues.

- a. The phenotype of strains deleted for the gene controlled by Mar are almost certainly more extreme than the phenotype when Mar control is eliminated. The authors should try, where possible to deconvolute the phenotype, so that we can understand the Mar-specific contribution, or alternatively view it in its proper environment. How much of the phenotype is due to MarA, and how much to the other regulators with overlapping targets.

The experiment they perform is to delete the Mar box, determine the phenotype and then complement with plasmid expressed genes having either a truncated promoter (*marbox-*, missing everything upstream of -35) or intact (*marbox+*). This strategy has two issues: (1) removing all sequence upstream of -35 may have some other effect on promoter activity unrelated to MarA binding, (2) the *marbox* is also a regulatory site for SoxS and Rob, and so deleting the *marbox* would remove their contribution to regulation as well as that of MarA. Potential solutions would be (1) using Δ *soxs* Δ *rob* background to isolate MarA effects (see Pomposiello et al. 2003 PMID:14594836 for selectively activatable MarA/SoxS/Rob strains and Northern blot analysis), (2) scrambling/inverting/deleting the *marbox*, (3) using MarA mutant that is unable to bind DNA. The

issue of multiple regulators could also be addressed by showing that SoxS and Rob do not bind those marboxes.

b. For *miaF*, the authors in vitro transcription experiments (Fig. 3E) as well as in vivo results from a β -galactosidase transcriptional fusion (Fig. 3F) indicate that there are multiple promoters, and that the mar box has only a 2-fold effect on transcription; yet in figure 3G and supps there is an 8-fold decrease in MIC when the mar box is removed. How do the authors reconcile these findings? If they believe the 2-fold change is indeed responsible, can they show this in a different way? Also, assuming the plasmid complementation vector has the other promoters, the Δ mar derivative should complement due to gene dosage unless P1 was also removed. Finally, permeability phenotypes of the Δ miaE and WT are shown, but there is no evidence to suggest that the lack of MarA regulation of that operon is enough to result in those phenotypes. In the absence of the marbox (which again could allow regulation by SoxS and Rob in addition to MarA) there is still transcription from the promoter (Fig 3F); this means that the Δ miaE phenotypes do not necessarily mimic the loss of MarA regulation, and are likely more extreme.

c. It may be visually helpful to use 2D-clustering of the genes and chemicals to find similarities, instead of sorting chemicals broadly by "target".

3. The authors cite the Ruiz & Levy (2010) that identified chromosomal mutations abrogating MarA multidrug resistance. Do the authors have any explanation for why was *mia* operon or *xseA* not identified in this way? Also, this work shows that many of the genes they identified altered MarA expression (i.e. they are required for MarA expression or stability). Do either *mia* or *xseA* mutations alter MarA expression?

Minor comments

a. Figure 1d: color bar is blue indicating greater fitness, which is the opposite of originally/commonly used. Consider changing color bar choice for clarity.

b. Line 100: "The data indicate that *xseA* is a determinant for MarA controlled quinolone tolerance." The data do not indicate that: the data indicate that *xseA* is a determinant for quinolone tolerance.

c. 28 peaks are near genes "shared" with *E. coli* K-12, however the authors do not clearly state what this means: are the sites (marboxes) themselves conserved or just the genes/orthologs that are nearby? Are the marboxes and spacings and potentially-regulated genes exactly the same? Clarifying this table legend (Table 1) is sufficient as they demonstrate binding for the K-12 regions they follow up on.

d. Lines 185-186 incorrectly reference panels of Fig 3.

e. Line 227: refers to "malFEDCB" instead of "miaFEDCB"

f. Not clear from the methods which *marR* plasmid used for ChIP (FLAG tagged or untagged protein)

Reviewer #3 (Remarks to the Author):

The study is premised on the observation that the *mar* phenotype may not be fully explained by MarA-dependent effect on drug permeability and efflux, i.e., MarA may activate other targets that contribute to intrinsic antibiotic resistance. The most straightforward way to identify such targets would be by comparing transcription profiles between *marA*⁺ and *marA*⁻ cells under relevant physiological conditions. Similar experiments aiming at defining MarA regulon have been done before but the discovered sets of genes had little in common.

The paper argues that newly discovered MarA targets contribute, in a measurable way, to the *mar* phenotype. The targets were discovered by ChIP-Seq and verified by molecular assays. However, despite this significant effort, the main claim of the paper that the "mar operon controls DNA repair and outer membrane integrity" remained just a hypothesis, albeit a plausible one. To prove this hypothesis, the authors have to demonstrate that transcription of *xseA* and *miaF-B* depends on MarA qualitatively and quantitatively in vivo. Specifically, one would have to demonstrate that:

1) when marA is inactivated, transcript abundances of the target genes go down; 2) when levels of MarA are up-modulated within a range of concentrations, levels of the target transcripts increase correspondingly, as should be evidenced by a respective regression model. The authors attempted to apply this approach in vitro in the case of mal promoter but the data in Figure 3 failed to establish a convincing quantitative association between the levels of MarA and of the target transcript. The complementation experiments used in lieu of an in vivo causative data are not sufficient, in part because they failed to establish that mar-box manipulations did not disturb the basal level of transcription of those genes. However, the effect observed in complementation experiments in the presence of ciprofloxacin is strong enough to be attributed to the full loss of function of the target genes, e.g., apparent susceptibility of the mar-less construct to 5 ng/ml in xseA transcription complementation experiment. If that is the case, then the MarA activity must be singularly responsible for the purported 40-fold MIC effect of xseA on the intrinsic antibiotic resistance. Yet, the increase of Cipro MIC in mar- mutants is only 2-6 -fold, and most of it is attributable to increased efflux and decreased intake of the drug. Thus one is forced to make a quantitative argument that MarA at its un-induced level controls most of xseA and mal transcription. Such pretty unexpected claim would require a much stronger evidence than the one presented in the paper.

Minor

Figure 2 and the accompanying explanation in the text are confusing. According to the figure, the xseA2 substring is where marbox is located, albeit mutated at the -36 position. However, a DNA fragment containing this sequence is not bound by MarA in the band-shift assay (2C, bottom). At the same time, according to the footprint coordinates (2D), the site labeled xseA2 is where DNA is somewhat protected by MarA from DNaseI digest. (Although it has to be noted that comparable in intensity protection and hypersensitivity patterns can be seen elsewhere along the footprint.)

Additionally, on line 108, authors state "As predicted, MarA bound to the xseA1 ..." Predicted how? The inferential analysis is missing from the paper and the Appendix could not be found.

P.3, ln.75: "The binding profiles of" what?

We thank the reviewers for taking the time to comment on our manuscript. We have taken the comments very seriously and have addressed the concerns experimentally or by modifying the text as appropriate. In some cases we would like to stand our ground and we hope the reviewers are able to consider the point of view we offer. Our responses are provided below.

Reviewers' comments:

Reviewer #1 (Remarks to the Author):

The manuscript by Sharma et al describes results using ChIP-seq to identify MarA targets in enterotoxigenic *Escherichia coli* strain H10407. Subsequent experiments using in vitro DNA binding and transcription assays and in vivo antibiotic sensitivity and promoter reporter assays in *E. coli* K12 strains reveal two new targets of MarA; *xseA* that functions in DNA repair and *mlaFEDCB* operon that functions in outer membrane integrity. These functions were also shown to be important for greater resistance to ciprofloxacin and deoxycline, respectively. Although several previous studies have attempted to identify the MarA regulon using transcriptomics, these earlier results did not lead to a common consensus regulon and direct effects of MarA were not established. Here the new results show that MarA can directly bind many promoter regions and can mediate its effect of multiple antibiotic resistance by activating two new target genes.

The strength of the paper is discovering that MarA activates expression of *xseA* and the *mla* operon, which increased resistance to certain antibiotics. Thus these results provide new insights into mechanisms of antibiotic resistance and the broader role of MarA. Overall, the authors present a compelling story but there are a few areas that require clarification.

1. The authors show that MarA directly binds these two promoter regions and that MarA can activate transcription of the P2 promoter of *mla*. To demonstrate that MarA also regulated these promoters in vivo, the authors compared *lacZ* expression controlled by the upstream control region to one where the upstream control region was truncated to remove the MarA binding site. However since the authors also mentioned that these binding sites are the same as those bound by SoxS, Rob, and RamA, how did the authors rule out a role for these other transcription factors?

We do not wish to rule out the possibility that the MarA targets will also be targets for SoxS and/or Rob (note RamA is not encoded by *E. coli*). Indeed, we think this is very likely and noted this in the discussion (“we suggest that MarA targets described here will bind closely related proteins”). Our conclusion is simply that genes reported are direct regulatory targets for MarA. If, as one would expect, the genes can also be regulated by SoxS or Rob this in no way alters our interpretation. We have however included extra data on this issue (see response to Reviewer 2). For example, we now show that increasing the level of MarA in the cell induces both *mlaFEDCB* and *xseA*. Reassuringly, such induction is marbox dependent.

This concern is reinforced by the fact that the ChIP-seq experiments were carried out in the enterotoxigenic *E. coli* strain but all of the in vivo assays were carried out with *E. coli* K12. Are the relevant promoter regions similar between these two types of *E. coli*?

The *xseA/mlaFEDCB* genes and intergenic regions are identical in *E. coli* K-12 and ETEC. This is also the case for the majority of MarA targets we identified. In the small number of cases where differences do occur these are limited to one or two single base changes in regulatory regions. Furthermore, these differences are never within the marbox sequence (see table below for summary). Similarly, deletions or insertions never occur. Hence, juxtaposition of regulatory elements is also conserved in all cases. We have added all of these

details to a new section in the supplementary methods in a section “Identification of H10407 MarA targets shared with K-12”. The information is also summarised below.

MarA target	% nucleotide identity (ETEC vs K-12)		
	gene	upstream 200bp	marbox
thrL	100%	100%	100%
leuL/leuO	100%	100%	100%
degP	99.9%	100%	100%
lacZ	99.9%	99.5%	100%
ybaO	100%	100%	100%
pheP	99.9%	100%	100%
modE<>acrZ	100%	100%	100%
ybiV	100%	100%	100%
grxA<>ybjC	100%	100%	100%
ycgF<>ycgZ	100%	99.5%	100%
fnr	100%	99.5%	100%
yneO	99.9%	100%	100%
marR	98%	99%	100%
yeeF	99%	100%	100%
ompC<>micF	99.9%	100%	100%
ypeC	97%	99.5%	100%
yfeS>>cysM	n.a.	100%*	100%
guaB<>xseA	100%	100%	100%
tolC	99.9%	100%	100%
(yhbV)	100%	n.a.	100%
miaF	100%	100%	100%
ibpA<>yidQ	100%	100%	100%
mnmG	100%	100%	100%
(yihT)	97%	n.a.	100%
(yiiG)	99%	n.a.	100%
yjcB<>yjcC	98%	100%	100%
yjjP<>yjjQ	99%	100%	100%
deoB	98%	99%	100%

*in this instance we refer to the entire region between convergent genes.

<> between divergent genes

>> between convergent genes

() within gene

Were either *xseA* or the *mia* operon induced by MarA in the previous transcriptomic studies?

Not in the Barbosa and Levy paper, but neither were the majority of known MarA targets in Ecocyc. For Bruce Demple’s paper we have no way to know; the full data are no longer available online and the manuscript does not list all differentially regulated genes.

Unfortunately, these papers came out before submitting to a database was commonplace so it’s difficult to check.

2. The authors have concluded that resistance to deoxycline can be attributed to the action of MarA on the *miaFEDCB* operon. Furthermore, the authors have shown that *miaF* can be transcribed from three promoter elements (P1, P2 and P3). Based on the location of the marbox within the promoter region, they conclude that *miaF* is transcribed primarily from P2.

This is not quite what we conclude. Our conclusion is that P2 is the MarA regulated promoter. In the absence of MarA activation, we do not think P2 is the major promoter.

However the truncation of this promoter that they use to demonstrate a dependence on MarA also has the upstream promoter P1 deleted. Thus it is unclear how observed changes in expression can be concluded to be exclusively due to the action of MarA.

We agree with the reviewer; this is an important issue to resolve. To avoid confounding effects of P1, we present a new set of experiments. Briefly, we have made two new DNA fragments named *m1aF1.1* and *m1aF2.1*. These are shown below alongside the starting *m1aF1* sequence. In both new fragments, the P1 promoter has been inactivated by mutating two key bases in the -10 hexamer (highlighted by red box). Hence, because P1 is already inactivated, the effect of deleting the marbox can be measured independently of any P1 associated effects. The result confirms that deletion of the marbox, rather than P1, leads to a reduction in activity. The data are shown in Figure S6 and below. As an aside, the new result shows a much clearer MarA effect. Presumably, this is because the overlapping P1 and P2 promoters normally interfere with each other.

Figure S6: Effect of marbox deletion following *m1aFP1* inactivation

The new data describe derivatives of the *m1aFEDCB* regulatory region that lack the P1 promoter due to point mutations in the -10 hexamer (DNA sequences *m1aF1.1* and *m1aF1.2*). These sequences have been fused to *lacZ* in plasmid pRW50 and *lacZ* expression has been determined in JCB387 cells. Deletion of the marbox still has a clear effect.

3. Previous studies have found that dmsX mediated activation of marA leads to multi drug resistance by ArcAB-TolC efflux pumps. However, the authors suggest an alternative explanation of multi-drug resistance but do not explain the discrepancy in results between these two studies.

We do not know specifically which paper is being referred to and cannot compare data with our own. However, we understand the question being asked. As the reviewer notes, previous studies attribute MDR mediated by MarA solely to AcrAB-TolC. Our argument is that MarA mediated MDR is more complicated; systems such as *m1aFEDCB* and *xseA* are also very important and widely conserved. We don't think there is a discrepancy. Rather, the change in dogma is that AcrAB-TolC is not the only MarA target that can mediate inherent antibiotic susceptibility.

Minor comments

1. It was not clear whether the levels of MarA play an important role in whether MarA exerts an effect at a given promoter and whether this is important in some of the discrepancies between previous genomic studies. Are the levels of MarA the same between the two E. coli strains?

We have no data on this issue but we suggest that they must be very similar given the promoter sequence similarities.

2. Apparently only a few of the 12 confirmed MarA targets of E. coli K12 and reported in Ecocyc were found in the enterotoxigenic E. coli. Do the authors have an explanation?

It is usual for ChIP-seq experiments to identify a subset of targets. This is because, under any given growth condition, other factors can occlude binding sites for the regulator of interest. The best exemplified case is for FNR, where 111 binding sites are blocked by H-NS, Fis or IHF (Meyers *et al.* 2013). This issue is particularly pertinent to MarA because of extensive competition with SoxS and Rob. Hence, *in vitro* binding constants suggest that most known marboxes are actually preferentially bound SoxS or Rob (Table S1). Consistent with this, the MarA ChIP-seq identifies known marboxes only if they bind MarA tightly *in vitro* (Table S1). It is also worth noting that the evidence for around half of “known” MarA targets is actually quite poor (see Table S1 for a summary).

3. Line 105. The spelling of electrophoretic.

Corrected.

4. Lines 115-122. Clarify whether it was previously known that xseA mutants are hypersensitive to ciprofloxacin.

This change has been made.

5. Lines 126-136. Clarify whether strains lacking XseB are also hypersensitive to ciprofloxacin.

The text has been altered. Strains lacking *xseB* are hypersensitive to ciprofloxacin.

What evidence allows you to conclude that XseA acts independently of XseB?

For clarification, we did not conclude that XseA acts independently of XseB. We considered this as one possible model and tested this by mutating the XseB interaction surface of XseA. On the basis of our data we conclude that XseA and XseB act together as a complex.

6. Line 256. The spelling of immunoprecipitated.

Corrected.

7. What is the mutation “-36C” in Figure 2b and what is its significance?

The -36C mutation is to ensure that the remaining portion of the marbox is inactivated in the *xseA2* fragment. We have added this information.

8. The legend for Fig 1 C does not describe adequately what the Venn diagram represents.

We have modified the text.

Reviewer #2 (Remarks to the Author):

The authors report a ChIP analysis in the pathogenic Escherichia coli strain H10407 of the binding sites of MarA: a major regulator driving naturally occurring multidrug resistance (MDR), likely hoping to find pathogen-specific roles for MarA. Finding only 1 such candidate, the authors follow up their hits in E. coli K-12. Interestingly, the authors recover only a small fraction of the validated E. coli Mar-box regulated genes in Ecocyc (last updated 2009, so likely incomplete).

We would like to address two of the comments made above:

1) “Interestingly, the authors recover only a small fraction of the validated E. coli Mar-box regulated genes”: It is normal for ChIP-seq experiments to identify only a subset of potential binding targets for a given regulator. The primary reason is that many genuine binding sites

will be blocked by binding of other factors (often nucleoid associated proteins) in any given experimental condition. This is best demonstrated for FNR, where 111 of 187 targets were blocked by either Fis, IHF or H-NS (Myers *et al.* 2013. *PLoS Genet.* **9**:e1003565). This is very likely true for all regulators, including MarA. For instance, the marbox upstream of *acnA* overlaps with binding sites for CRP, FNR, and AcrA. Of course, if one considers competition with SoxS and Rob, it is easy to see how MarA could be excluded from certain targets. Importantly, *in vitro* binding constants, and gene expression assays, show that most known marboxes are primarily bound by SoxS or Rob (Table S1 below). Consistent with this, the MarA ChIP-seq only identifies known marboxes that are tight MarA binders and exhibit MarA regulation *in vivo* (Table S1). It is worth noting that experimental evidence for around half of the “known” MarA targets is actually quite poor (see Table S1 for a summary). For example, although they are considered targets, MarA binding to *pqiA*, *inaA*, *hdeA*, *sodA*, *fpr* and *purA* cannot be detected *in vitro* (Table S1). To support readers with interpretation of our ChIP-seq data, we have added a brief description of these issues to the Results section and a detailed description to Appendix S1 and Table S1.

Table S1: Interactions of MarA and SoxS with previously proposed targets

Gene ^a	MarA binding in vitro ^b	Affinity (Kd (nM) ⁻¹) ^c			Relative activation (SoxS/MarA) ^d	Other comments ^e
		MarA	SoxS	Rob		
micF	yes ⁷²	25	50	n.d.	1	
marRAB	yes ¹⁹	75	75	20	1.1	
ybiC	yes ¹⁹	320	100	35	0.8	
fumC	yes ¹⁹	320	75	75	5.3	
rob	yes ⁷³	400*	100*	<100*	n.d.	
nfsB	yes ⁷⁴	>500*	<200*	<200*	0.6	
acrAB	yes ¹⁹	800	128	35	1.8	
acnA	yes ¹⁹	1000	350	500	3.2	
zwf	yes ¹⁹	>1000	>1000	1000	2.3	
pqiA	none ¹⁹	n.d.	n.d.	n.d.	4	Martin et al. 2011 unable to detect binding of MarA or SoxS in vitro at concentrations tested.
inaA	none ¹⁹	n.d.	n.d.	n.d.	0.8	Martin et al. 2011 unable to detect binding of MarA or SoxS in vitro at concentrations tested.
hdeA	none ⁷⁵	n.d.	n.d.	n.d.	n.d.	Schneiders et al. 2004 identified a potential MarA site but binding could not be detected in vitro .
sodA	none ¹⁹	n.d.	n.d.	n.d.	2.3	Martin et al. 2011 unable to detect binding of MarA or SoxS in vitro at concentrations tested.
fpr	none ¹⁹	n.d.	200	n.d.	23	Martin et al. 2011 detected binding of SoxS but not MarA in vitro at concentrations tested.
purA	yes ⁷⁵	n.d.	n.d.	n.d.	n.d.	Schneiders et al. 2004 detected MarA binding in vitro but no comparison with SoxS was made

^aGenes listed are those described in Ecocyc as being MarA targets on the basis of high quality evidence (usually *in vitro* DNA binding assays). Predicted targets listed by Ecocyc are not included. Underlined genes were identified by ChIP-seq as MarA binding targets in this work. Genes in bold were identified by ChIP-exo as SoxS targets⁴⁷.

^bExperimental evidence for purified MarA binding to the proposed target. Numbers refer to references in the main body of the paper.

^cAffinities are reported where binding of MarA and SoxS were directly compared in the same paper. An asterisk indicates values estimated from published images of gel electrophoretic mobility shift assays rather than values directly provided by the authors. In the table, the order in which genes is based in affinity for MarA *in vitro*. If a binding constant could not be determined this is shown as n.d.

^dMartin *et al.* 2011 directly compared the ability of MarA and SoxS to activate transcription of the respective genes. A relative activation of >1 indicates more efficient activation by SoxS. If the comparison has not been done this is shown as n.d.

^eIn some instances Ecocyc lists genes as being MarA targets on the basis of DNA binding data. On inspection, DNA binding experiments were indeed done but no binding was detected under the conditions tested

2) “Ecocyc (last updated 2009, so likely incomplete)” We think this statement is mistaken. The last published list of Ecocyc updates was described in 2017 by Keseler *et al.* (*Nucleic Acids Res.* **45**:D543-50). We have also emailed Ecocyc and have been assured that the database is continually updated. Indeed, examination of the Ecocyc page for MarA shows that publications from as recently as 2017 are included (<https://ecocyc.org/gene?orgid=ECOLI&id=PD00365>). Searching PubMed using the term “MarA coli” found no relevant papers not present in Ecocyc. Briefly, the last attempt to identify new MarA target genes was Bob Martin’s 2011 Mol Micro paper. All of these targets are in Ecocyc.

Although this is an incomplete global study, the authors do characterize two new Mar targets *xseA* (encoding a subunit of Exonuclease VII) and the phospholipid transfer system encoded by *mlaFEDCB* (previously identified as a target in *Pseudomonas*). This manuscript would be greatly improved by:

(1) greater transparency about the existing literature on marboxes and MarA-confirmed sites: As noted above, we believe that we do cover the existing literature fully. Ecocyc is up-to-date

and a search of PubMed reveals no papers that we or EcoCyc have missed. To aid transparency, we have included Table S1 (above) that describes the existing evidence more completely and cites relevant publications.

(2) discussion of H10407-found MarA binding sites vs K-12 Mar boxes: This has been dealt with in our Response to Reviewer 1 and we further expand on this in our responses below. Extra details have also been added to the Appendix.

(3) evidence for the specific role of MarA at the defined sites (as opposed to the combined regulation of MarA/SoxS/Rob): In our manuscript, we were very careful not to exclude the possibility of overlapping regulation by SoxS or Rob. Indeed, we explicitly stated in the discussion that “MarA targets described here will bind closely related proteins”. This is to be expected and is a general feature of the mar/sox/rob regulon. Hence, while noting possible SoxS/Rob overlap, we conclude that the marboxes identified do bind MarA in our conditions (demonstrated by direct MarA binding *in vivo* and *in vitro*) and that the marbox is regulatory (demonstrated by *in vitro* transcription and *lacZ* fusion assays). We think these conclusions are reasonable. Even so, we have taken the reviewer’s comments seriously and have done experiments to better understand specificity. In the revision we i) demonstrate a correlation between intracellular MarA levels and activity of the defined promoters ii) show that this correlation requires the marbox iii) check specificity using DNA binding assays as suggested by the reviewer (see below for further details).

Major concerns

1. The authors seek to comprehensively map the regulon of MarR and MarA, in the pathogenic *E. coli* strain H10407, using ChIP using a marA or marR overexpression plasmid. Even though overexpression may overestimate binding sites, there is no evidence that their approach is anywhere near comprehensive. In this regard, it is important to determine whether the inability to identify previously validated *E. coli* Mar-controlled proteins derives from the fact that H10407 lacks Mar binding sites upstream of these genes or whether their approach was unable to detect these sites.

As noted above, ChIP-seq never identifies all binding sites; other DNA binding proteins simply block access to many targets. Hence, we did not seek or claim to *comprehensively* map the MarA regulon. Rather, we sought to *accurately* map binding of MarA in the conditions of our experiment. We believe we have done this. The major benefit of ChIP-seq is that it identifies directly regulatory targets, albeit not all of them. Note that we also checked all binding sites by gel shift assays and only one did not bind purified MarA *in vitro*.

This analysis will not only indicate whether there results are likely to be comprehensive; it will also indicate whether there is a great deal of variability in the Mar regulon between closely related organisms. At the very least, the authors could assess bioinformatically whether validated *E. coli* targets have Mar boxes in H10407.

We apologise for not being more specific about this issue and have done the bioinformatics. The MarA regulon is extremely similar between closely related organisms (e.g. different *E. coli* strains). This is exemplified in a table provided above in the response to Reviewer 1. To clarify, the reason we do not see certain targets using ChIP-seq is not because they are absent in H10407. Rather, this is a normal aspect of ChIP-seq analysis. As noted, many sites will be masked by other DNA binding proteins (including SoxS and Rob).

Additionally, it would be helpful in Table 1 to show K-12 EcoCyc confirmed MarA regulated genes that were not found by the authors’ experiment.

This information is in the new Table S1. The new table makes it clear that the targets we don't detect have a low affinity for MarA *in vitro* and preferentially bind SoxS and Rob.

2. Phenotypic characterization of the gene deletion phenotypes are a nice addition to the ms. However, there are several issues.

a. The phenotype of strains deleted for the gene controlled by Mar are almost certainly more extreme than the phenotype when Mar control is eliminated.

We agree.

The authors should try, where possible to deconvolute the phenotype, so that we can understand the Mar-specific contribution, or alternatively view it in its proper environment.

We agree, and in some cases we did attempt to deconvolute the phenotype by deleting the marbox (i.e. we determined the marbox specific contribution). In the revision, we have added further experiments of this type (Figure S8).

How much of the phenotype is due to MarA, and how much to the other regulators with overlapping targets.

As noted above, we in no way exclude the possibility that SoxS or Rob could contribute in conditions where they are induced. However, under the conditions of our experiment, we know that MarA is bound to the marbox (from the ChIP-seq data) and that the marbox is required for promoter activity. As outlined elsewhere in our responses, we have further defined specificity using MarA overexpression assays (see response to Reviewer 3) and DNA binding assays (see below). This should aid interpretation of the phenotypic data.

The experiment they perform is to delete the Mar box, determine the phenotype and then complement with plasmid expressed genes having either a truncated promoter (marbox-, missing everything upstream of -35) or intact (marbox+). This strategy has two issues: (1) removing all sequence upstream of -35 may have some other effect on promoter activity unrelated to MarA binding

The reviewer is correct; other sequences upstream of -35 may have an effect. Reviewer 1 made a similar comment. As described in our response to Reviewer 1, we have included a new set of constructs where potential upstream DNA effects have been negated for the *mfaFEDCB* experiments (Figure S6). Furthermore, for both *xseA* and *mfaFEDCB*, we have shown that gradually increasing the amount of MarA in the cell (using IPTG inducible *marA*) leads to a gradual increase in promoter activity. Such increases are lost when the marbox is removed (Figure S3). Also, note that the *in vitro* transcription assays for *mfaFEDCB* clearly demonstrate direct activation. We hope this convinces the reviewer that MarA does activate expression of these genes.

The marbox is also a regulatory site for SoxS and Rob, and so deleting the marbox would remove their contribution to regulation as well as that of MarA. Potential solutions would be

We agree and again stress our explicit statement that regulators related to MarA will likely bind the identified targets. However, we have done two of the three experiments suggested.

(1) using $\Delta\text{soxs}\Delta\text{rob}$ background to isolate MarA effects (see Pomposiello et al. 2003 PMID:14594836 for selectively activatable MarA/SoxS/Rob strains and Northern blot analysis) We appreciate the reviewer's comment and have done an experiment very similar to that suggested. Briefly, we constructed a plasmid encoding selectively activatable *marA* under the control of an IPTG inducible promoter. We show that, in the presence of this plasmid, addition of IPTG increases expression from the *mfaFEDCB* and *xseA* promoters.

Importantly, this activation requires the marbox. Hence, MarA must be directly activating these genes *in vivo* via the site we identify. We did the experiment in wild type cells, rather than a $\Delta soxS\Delta rob$, since this provides evidence that SoxS and Rob are unable to block activation by MarA should they be competing for the same site.

(2) scrambling/inverting/deleting the marbox, we don't understand how the suggested experiment differs to those already presented (i.e. the reviewer suggests deleting the marbox but we have done this). Also, we do not think scrambling or inverting the marbox would address the issue of specificity (i.e. this would surely also affect SoxS and Rob binding).

(3) using MarA mutant that is unable to bind DNA. The issue of multiple regulators could also be addressed by showing that SoxS and Rob do not bind those marboxes. We agree, binding assays would be informative and these experiments have been done. However, there is a major complication associated with this experiment that the reviewer does not mention. Briefly, it is well established that Rob binds to all marboxes much more tightly than MarA or SoxS (e.g. Martin *et al.* 2002. *Mol Micro.* **43**:355-370 & Kwon *et al.* *Nat. Struc. Biol.* 2000. **7**:424-430). This includes marboxes where MarA is the primary activator (e.g. *tolC*, *acrAB* and *marRAB*). For example, the affinity of Rob for the *acrAB* promoter is 22-fold higher than for MarA (Martin *et al.* 2002). A further complication is that Rob binds DNA non-specifically in the absence of a marbox (Martin *et al.* 2002). Indeed, Rob often has a higher affinity for non-specific DNA than MarA does for genuine marboxes (Martin *et al.* 2002 and our own data). The observation that Rob behaves in this way, combined with the fact that Rob is constitutively expressed at ~5,000 copies per cell, appears paradoxical. The “Rob paradox” is discussed at length by Martin *et al.* (2002) and elsewhere. It is proposed that sequestration of Rob in foci, and the much tighter binding of SoxS and MarA to RNA polymerase, counteract Rob's higher affinity for DNA. Consistent with previous reports, we also find that Rob binds DNA tightly but with low sequence specificity. Hence, Rob binding to a control DNA fragment, containing no marbox, is evident (Figure S9a). Indeed, the affinity of Rob for the control DNA fragment was similar to the affinity of MarA for the *marRAB* promoter (compare Figure S9 a and S9b). Furthermore, Rob bound to the *marRAB* promoter with 5-fold higher affinity than MarA (Figure S9b).

Figure S9: Rob binds DNA with high affinity but low specificity

Given the unusual DNA binding properties of Rob (i.e. that it binds tightly to any DNA sequence) we can deduce little from Rob DNA binding assays. However, DNA binding assays do address relative specificity of SoxS and MarA. Our data show that both the *xseA* and *mfaFEDCB* promoters preferentially bind MarA rather than SoxS (Figure S10). Our findings are consistent with ChIP-exo experiments that detect no binding of SoxS at *mfaFEDCB* or *xseA* (Seo *et al.* 2015. *Cell Reports* **12**:1289-1299). **As the situation is complicated, we have included all of the new data, as well as associated introductory**

and discursive text, in the appendix. It is not possible to cover the topic fully within the word limits for the main text.

Figure S10: The *mfaFEDCB* and *xseA* regulatory regions preferentially bind MarA rather than SoxS

The figure shows binding of proteins (0.4, 1.2 or 2.0 μM) to the a) *mfaFEDCB* or b) *xseA* regulatory DNA regions.

b. For *mfaF*, the authors in vitro transcription experiments (Fig. 3E) as well as in vivo results from a β -galactosidase transcriptional fusion (Fig. 3F) indicate that there are multiple promoters, and that the mar box has only a 2-fold effect on transcription; yet in figure 3G and sups there is an 8-fold decrease in MIC when the mar box is removed. How do the authors reconcile these findings? If they believe the 2-fold change is indeed responsible, can they show this in a different way?

The reviewer states “there is an 8-fold decrease in MIC when the marbox is removed”. We think the reviewer has misunderstood the data presented. Briefly, Figure 3G does not report MIC changes. Rather, the figure shows differences in growth at a single antibiotic concentration. Similarly, although Table S1 does report MIC values, the changes are due to deletion of *mfaE*, not deletion of the marbox. Thus, the 8-fold effect is due to the deletion of *mfaE*. As the reviewer comments below “ $\Delta mfaE$ phenotypes do not necessarily mimic the loss of MarA regulation”. Hence, by the reviewers own logic, we can see no contradiction in our data. As an aside, when quantified, MarA activates P2 by 7-fold *in vitro*. Furthermore, an 8-fold stimulatory effect of the marbox on P2 is evident *in vivo* if P1 is first inactivated (Figure S6).

Also, assuming the plasmid complementation vector has the other promoters, the Δ mar derivative should complement due to gene dosage unless P1 was also removed.

The P1 promoter was removed along with the marbox since the two overlap (the P1 -35 element is within the marbox).

Finally, permeability phenotypes of the Δ mleA and WT are shown, but there is no evidence to suggest that the lack of MarA regulation of that operon is enough to result in those phenotypes. In the absence of the marbox (which again could allow regulation by SoxS and Rob in addition to MarA) there is still transcription from the promoter (Fig 3F); this means that the Δ mleA phenotypes do not necessarily mimic the loss of MarA regulation, and are likely more extreme.

We did not claim that “deletion” and “loss of regulation” phenotypes would be the same. The purpose of the deletion experiments was to test the hypothesis that the *mle* system was altering surface hydrophobicity and so drug uptake. We believe our data strongly support this hypothesis. Even so, we have repeated the experiments to also determine the “loss of marbox phenotype”. As the reviewer suggests, the “loss of regulation” phenotype is slightly less severe than the “deletion” phenotype (Figure S8). Nevertheless, there is still a clear effect and the new data fully support our model.

Figure S8: Effect of *mleFEDCB* marbox deletion on doxycycline uptake and cell surface hydrophobicity

The data are equivalent to those in Figures 3h, 3j and 3k except that we used the Δ mleA strain transformed with pBR322 encoding *mleFEDCB* with or without the upstream marbox.

c. It may be visually helpful to use 2D-clustering of the genes and chemicals to find similarities, instead of sorting chemicals broadly by “target”.

We are not sure exactly what is meant by “2D-clustering of the genes and chemicals”. If the reviewer can clarify their request we will happily consider this.

3. The authors cite the Ruiz & Levy (2010) that identified chromosomal mutations abrogating MarA multidrug resistance. Do the authors have any explanation for why was *mle* operon or *xseA* not identified in this way? Also, this work shows that many of the genes they identified altered MarA expression (i.e. they are required for MarA expression or stability). Do either *mle* or *xseA* mutations alter MarA expression?

There is an obvious explanation; Ruiz & Levy did not directly screen for mutations abrogating MarA mediated multidrug resistance. As Ruiz & Levy note, “we would obtain

many mutants that affected MDR by mechanisms unrelated to MarA”. Instead, Ruiz & Levy screened for chromosomal mutations that prevented repression of the *hdeAB* promoter by MarA. There is no reason to think that *mfaFEDCB* or *xseA* mutation would alter *hdeAB* repression. The same is true of most MarA regulated genes (also not found by the screen). In fact, as turned out to be the case, one would expect such a screen to isolate i) *hdeAB* specific regulators or ii) mutations that alter MarA levels.

Minor comments

a. Figure 1d: color bar is blue indicating greater fitness, which is the opposite of originally/commonly used. Consider changing color bar choice for clarity.

We chose the colour scheme to fit with the rest of the paper and would like to keep as is.

b. Line 100: “The data indicate that *xseA* is a determinant for MarA controlled quinolone tolerance.” The data do not indicate that: the data indicate that *xseA* is a determinant for quinolone tolerance.

We have altered the text.

c. 28 peaks are near genes “shared” with E. coli K-12, however the authors do not clearly state what this means: are the sites (marboxes) themselves conserved or just the genes/orthologs that are nearby? Are the marboxes and spacings and potentially-regulated genes exactly the same? Clarifying this table legend (Table 1) is sufficient as they demonstrate binding for the K-12 regions they follow up on.

We have clarified this in the text. Briefly, everything is shared. The marboxes are all identical in sequence and the spacing with respect to other promoter elements is unchanged. The genes are also identical or extremely similar (>97% nucleotide identity). We have added details to the Appendix S1.

d. Lines 185-186 incorrectly reference panels of Fig 3.

This has been corrected.

e. Line 227: refers to “*malFEDCB*” instead of “*mfaFEDCB*”

This has been corrected.

f. Not clear from the methods which marR plasmid used for ChIP (FLAG tagged or untagged protein)

These details are provided in the supplementary methods in Appendix SI.

Reviewer #3 (Remarks to the Author):

The study is premised on the observation that the mar phenotype may not be fully explained by MarA-dependent effect on drug permeability and efflux, i.e., MarA may activate other targets that contribute to intrinsic antibiotic resistance. The most straightforward way to identify such targets would be by comparing transcription profiles between marA⁺ and marA⁻ cells under relevant physiological conditions. Similar experiments aiming at defining MarA regulon have been done before but the discovered sets of genes had little in common. The paper argues that newly discovered MarA targets contribute, in a measurable way, to the mar phenotype. The targets were discovered by ChIP-Seq and verified by molecular assays. However, despite this significant effort, the main claim of the paper that the “mar operon controls DNA repair and outer membrane integrity” remained just a hypothesis, albeit a plausible one. To prove this hypothesis, the authors have to demonstrate that transcription of *xseA* and *mfaF-B* depends on MarA qualitatively and quantitatively *in vivo*. Specifically, one would have to demonstrate that: 1) when marA is inactivated, transcript abundances of the

target genes go down; 2) when levels of MarA are up-modulated within a range of concentrations, levels of the target transcripts increase correspondingly, as should be evidenced by a respective regression model. The authors attempted to apply this approach *in vitro* in the case of *mal* promoter but the data in Figure 3 failed to establish a convincing quantitative association between the levels of MarA and of the target transcript. The complementation experiments used in lieu of an *in vivo* causative data are not sufficient, in part because they failed to establish that *mar*-box manipulations did not disturb the basal level of transcription of those genes.

We appreciate the comment and have done the experiment requested by the reviewer. To examine the response of promoters *in vivo*, to differing levels of MarA, we made a plasmid encoding *marA* under the control of an IPTG inducible promoter. Our data show that increasing the concentration of IPTG in the growth media leads to an increase in promoter activity for both *mfaFEDCB* and *xseA*. Furthermore, when these measurements are repeated with promoter derivatives lacking the *mar*box, the IPTG effect is lost (Figure S3). We hope the reviewer finds this a compelling demonstration of MarA activation *in vivo*. The reviewer suggests that “when *marA* is inactivated, transcript abundances of the target genes go down”. This is often not the case because SoxS and Rob compensate for loss of MarA (see Zhang *et al.* 2008. *Mol Micro.* **69**:1450-1455; Chubiz & Rao. 2011. *J. Bacteriol.* **193**:2252-2260). It is for precisely this reason that most MarA regulated genes cannot be identified by comparing *marA*⁺ and *marA*⁻ transcription profiles. This is why we instead compared the effect of overproducing MarA in the presence and absence of the *mar*box. We think this is a more meaningful experiment. We have quantified the *in vitro* transcription data in Figure 3e. Activation of the P2 promoter by MarA is unequivocal; MarA stimulates activation 7-fold.

Figure S3: Increased expression of *mfaFEDCB* and *xseA* due to increased intracellular MarA requires the *mar*box

The figure shows β -galactosidase activities determined from lysates of strain T7 express carrying two plasmids. The first plasmid encodes *marA* under the control of an IPTG inducible promoter. The second plasmid is a pRW50 derivative encoding the *mfaFEDCB* or *xseA* promoter fused to *lacZ*.

However, the effect observed in complementation experiments in the presence of ciprofloxacin is strong enough to be attributed to the full loss of function of the target genes, e.g., apparent susceptibility of the *mar*-less construct to 5 ng/ml in *xseA* transcription complementation experiment. If that is the case, then the MarA activity must be singularly responsible for the purported 40-fold MIC effect of *xseA* on the intrinsic antibiotic resistance. Yet, the increase of Cipro MIC in *mar*- mutants is only 2-6 –fold, and most of it is attributable to increased efflux and decreased intake of the drug. Thus one is forced to make a quantitative argument that MarA at its un-induced level controls most of *xseA* and *mal* transcription. Such pretty unexpected claim would require a much stronger evidence than the one presented in the paper.

We are not certain of the point being made but have tried to interpret the reviewer’s comments and respond. We believe the reviewer is questioning the relative phenotypes

resulting from i) deletion of *xseA* ii) deletion of the marbox upstream of *xseA* and iii) deletion of *marA*. The reviewer notes that reported changes in sensitivity to ciprofloxacin, resulting from deletion of *marA*, are smaller than the effect we report due to deletion of *xseA* or the upstream marbox. We do not think this is surprising. It seems very unlikely that *marA* mutants would totally lose the ability to express *xseA*. This is because there is redundancy in most gene regulatory systems and this is particularly true for genes controlled by MarA; factors such as SoxS and Rob simply compensate for loss of MarA (Zhang *et al.* 2008. *Mol Micro.* **69**:1450-1455). Indeed, this is precisely why it has been impossible to define the MarA regulon using transcriptomics to compare wild type and $\Delta marA$ cells. Importantly, complete removal of *xseA* or the marbox removes the possibility of compensatory regulation and so is expected to have a bigger effect than deletion of *marA*. There is also an assumption that *marRAB* is uninduced in our experiments. However, we ask the reviewer to recall that antibiotics are always present in our assays. For example, all of the complementation experiments use cultures with 100 $\mu\text{g/ml}$ ampicillin and this is known to induce *marRAB* (Kaldalu *et al.* 2004. *Antimicrob. Agents Chemother.* **48**:890-896). Ciprofloxacin also induces *mar* according to RNA-seq analysis (Shishkin *et al.* 2015. *Nature Methods.* **12**:323–325) and there are many reports of *marRAB* being induced by tetracyclines (e.g. Ariza *et al.* 1994. *J. Bacteriol.* **176**:143-148). Furthermore, even in non-inducing conditions, there are ~200 molecules of MarA per cell (Matrin *et al.* 2002. *Mol Micro.* **43**:355-370).

Minor

Figure 2 and the accompanying explanation in the text are confusing. According to the figure, the *xseA2* substring is where marbox is located, albeit mutated at the -36 position. However, a DNA fragment containing this sequence is not bound by MarA in the band-shift assay (2C, bottom).

We are happy to clarify; the marbox is not located within *xseA2*. The text states that the 5' end of the *xseA1* and *xseA2* DNA fragments is shown by the inverted triangle in Figure 2b. Hence, in the *xseA2* fragment, half of the marbox has been “chopped off”. This is reiterated in the figure legend. The -36 mutation inactivates the remaining portion of the marbox in the *xseA2* fragment. We have modified the text and hope it is now easier to follow.

At the same time, according to the footprint coordinates (2D), the site labeled *xseA2* is where DNA is somewhat protected by MarA from DNaseI digest. (Although it has to be noted that comparable in intensity protection and hypersensitivity patterns can be seen elsewhere along the footprint.)

Yes, this is expected. As explained above, the 5' end of the *xseA2* fragment falls within the marbox.

Additionally, on line 108, authors state “As predicted, MarA bound to the *xseA1* ...” Predicted how? The inferential analysis is missing from the paper and the Appendix could not be found.

Predicted on the basis that *xseA1* contains a marbox whilst *xseA2* does not.

P.3, ln.75: “The binding profiles of” what?

We have added further information.

REVIEWERS' COMMENTS:

Reviewer #1 (Remarks to the Author):

The authors have adequately responded to and addressed the reviewers comments.

Reviewer #2 (Remarks to the Author):

We appreciate the authors' efforts to clarify their findings and to address our questions. Overall we find that clarity is improved based on (1) the data presented in Supplementary Methods (comparison of H10407 and K-12 sites) and Table S1 (explanation of previously reported MarA-regulated sites), and (2) the manuscript text and new experiments which now explicitly address our concern about MarA vs. SoxS/Rob regulation.

The text of the manuscript now more clearly states the expectations of the ChIP-seq experiment to determine the mar regulon (lines 93-96): in our opinion this no longer can be misinterpreted as "comprehensive". We appreciate the authors' explanation of the EcoCyc-annotated MarA sites as well, and are gratified to see a detailed explanation for why certain sites were not recovered in their experiment, now in Table S1. The logical connection between the experiment in H10407 and the follow up in K-12 is now also made clear by the information in the Supplementary Methods/Appendix, showing the high degree of similarity of those sites across these two strains.

The authors have provided two additional experiments to address the possible confounding effects of disrupting MarA regulation and SoxS/Rob compensatory regulation. The first is to demonstrate that increasing MarA concentration in the cell (IPTG-inducible marA construct) increases transcription from promoters of the identified targets xseA and mlaFEDCB in an IPTG-dependent manner; this was measured using lacZ fusions to the identified promoters (plasmid-expressed), and in both cases is dependent on the promoters' marboxes (Figure S3). As the authors note: in the absence of MarA-dependent regulation, SoxS/Rob may compensate, and this experiment avoids that issue.

The second new experiment shows in vitro affinities of MarA, SoxS, and Rob for the identified MarA targets (Figure S10), demonstrating that Rob has low specificity for tightly binding DNA (Figure S9), and that MarA has a higher affinity for the identified targets than SoxS. This aligns with the in vivo finding that overexpressing MarA increases transcription at those sites, and suggests that MarA is the dominant regulator at those sites.

The contribution of MarA-dependent regulation to drug uptake / surface hydrophobicity via mlaE expression is also clarified in the revised text. The combination of Figure 3H and Figure S8A now demonstrates a continuum of doxycycline-accumulation phenotypes from (greatest accumulation to least accumulation): Δ mlaE, Δ mlaE + complemented P(Δ marbox)-mlaE, Δ mlaE + complemented mlaE, WT.

We wish to briefly clarify our suggestions for presenting the chemical phenotype data in Figure 1D that may help to maximize the visual information for the reader. Hierarchical clustering of both the genes and the chemicals (two-dimensional clustering) is an unbiased way to identify patterns in the data, by simultaneously sorting genes and chemicals based on similarity. The result is a genes x chemicals grid where similar signals are emphasized. Alternatively, if the "Antibiotic target" categories are important for demonstrating the diversity of resistances potentially controlled by MarA (which is a valid reason), the authors could hierarchically cluster the genes (one-dimensional clustering). The result of that analysis would be to show that some MarA targeted genes have more similar chemical profiles than others. Although not necessary, this analysis is simple to do provides a small extra layer of visual clarity for the reader. Cluster 3 is a straightforward software to implement this analysis (<http://bonsai.hgc.jp/~mdehoon/software/cluster/software.htm>).

The authors provide a new analysis of the conservation of their MarA-regulated sites across the Enterobacteriaceae, including genomes where the regulated genes are present but the marboxes are mutated/absent. The most extensive conservation of MarA-regulated sites occurs within closely-related Escherichia and Shigella (relationships shown in cladogram), however some regulated sites are conserved further out (ex. in Salmonella and Raoultella). Notably, mlaF regulation by this analysis is more extensively conserved than xseA regulation. This analysis raises interesting questions about the evolution of the mar regulon.

Minor corrections:

- line 190: "Gram negative" should be "Gram-negative"
- line 106: "teracyclines" should be "tetracyclines"

Reviewer #3 (Remarks to the Author):

Thank you for your clarifications.

We thank the reviewers for assessing the revised version of our paper and we are glad that they found it to be acceptable. Reviewers 1 and 2 had no further comments. Reviewer 3 gave a lengthy description of the changes we made but raised only three additional points. Two of these were minor typographical errors that we have corrected. The third was a description of the 2-D clustering approach that they suggested may be useful for Figure 1d. Although the reviewer noted that this was not necessary, we did look at the suggestion in more detail and tried the Cluster 3 software that they suggested. However, we found that this did not improve clarity and so have not modified Figure 1d.

Reviewer #1 (Remarks to the Author):

The authors have adequately responded to and addressed the reviewers comments.

Reviewer #2 (Remarks to the Author):

We appreciate the authors' efforts to clarify their findings and to address our questions. Overall we find that clarity is improved based on (1) the data presented in Supplementary Methods (comparison of H10407 and K-12 sites) and Table S1 (explanation of previously reported MarA-regulated sites), and (2) the manuscript text and new experiments which now explicitly address our concern about MarA vs. SoxS/Rob regulation.

The text of the manuscript now more clearly states the expectations of the ChIP-seq experiment to determine the mar regulon (lines 93-96): in our opinion this no longer can be misinterpreted as "comprehensive". We appreciate the authors' explanation of the EcoCyc-annotated MarA sites as well, and are gratified to see a detailed explanation for why certain sites were not recovered in their experiment, now in Table S1. The logical connection between the experiment in H10407 and the follow up in K-12 is now also made clear by the information in the Supplementary Methods/Appendix, showing the high degree of similarity of those sites across these two strains.

The authors have provided two additional experiments to address the possible confounding effects of disrupting MarA regulation and SoxS/Rob compensatory regulation. The first is to demonstrate that increasing MarA concentration in the cell (IPTG-inducible marA construct) increases transcription from promoters of the identified targets xseA and mlaFEDCB in an IPTG-dependent manner; this was measured using lacZ fusions to the identified promoters (plasmid-expressed), and in both cases is dependent on the promoters' marboxes (Figure S3). As the authors note: in the absence of MarA-dependent regulation, SoxS/Rob may compensate, and this experiment avoids that issue.

The second new experiment shows in vitro affinities of MarA, SoxS, and Rob for the identified MarA targets (Figure S10), demonstrating that Rob has low specificity for tightly binding DNA (Figure S9), and that MarA has a higher affinity for the identified targets than SoxS. This aligns with the in vivo finding that overexpressing MarA increases transcription at those sites, and suggests that MarA is the dominant regulator at those sites.

The contribution of MarA-dependent regulation to drug uptake / surface hydrophobicity via mlaE expression is also clarified in the revised text. The combination of Figure 3H and Figure S8A now demonstrates a continuum of doxycycline-accumulation phenotypes from (greatest accumulation to

least accumulation): $\Delta mlaE$, $\Delta mlaE$ + complemented P($\Delta marbox$)- $m laE$, $\Delta mlaE$ + complemented $m laE$, WT.

We wish to briefly clarify our suggestions for presenting the chemical phenotype data in Figure 1D that may help to maximize the visual information for the reader. Hierarchical clustering of both the genes and the chemicals (two-dimensional clustering) is an unbiased way to identify patterns in the data, by simultaneously sorting genes and chemicals based on similarity. The result is a genes x chemicals grid where similar signals are emphasized. Alternatively, if the “Antibiotic target” categories are important for demonstrating the diversity of resistances potentially controlled by MarA (which is a valid reason), the authors could hierarchically cluster the genes (one-dimensional clustering). The result of that analysis would be to show that some MarA targeted genes have more similar chemical profiles than others. Although not necessary, this analysis is simple to do provides a small extra layer of visual clarity for the reader. Cluster 3 is a straightforward software to implement this analysis (<http://bonsai.hgc.jp/~mdehoon/software/cluster/software.htm>).

The authors provide a new analysis of the conservation of their MarA-regulated sites across the Enterobacteriaceae, including genomes where the regulated genes are present but the marboxes are mutated/absent. The most extensive conservation of MarA-regulated sites occurs within closely-related Escherichia and Shigella (relationships shown in cladogram), however some regulated sites are conserved further out (ex. in Salmonella and Raoultella). Notably, $m laF$ regulation by this analysis is more extensively conserved than $x seA$ regulation. This analysis raises interesting questions about the evolution of the mar regulon.

Minor corrections:

- line 190: “Gram negative” should be “Gram-negative”
- line 106: “teracyclines” should be “tetracyclines”

Reviewer #3 (Remarks to the Author):

Thank you for your clarifications.